# R-loop formation during S phase is restricted by PrimPol-mediated repriming

Saša Šviković[1], Alastair Crisp[1], Sue Mei Tan-Wong[2], Thomas A Guilliam[3], Aidan J Doherty[3], Nicholas J Proudfoot[2], Guillaume Guilbaud[1] & Julian E Sale[1,*] 🆔

## Abstract

**During DNA replication, conflicts with ongoing transcription are frequent and require careful management to avoid genetic instability. R-loops, three-stranded nucleic acid structures comprising a DNA:RNA hybrid and displaced single-stranded DNA, are important drivers of damage arising from such conflicts. How R-loops stall replication and the mechanisms that restrain their formation during S phase are incompletely understood. Here, we show *in vivo* how R-loop formation drives a short purine-rich repeat, (GAA)$_{10}$, to become a replication impediment that engages the repriming activity of the primase-polymerase PrimPol. Further, the absence of PrimPol leads to significantly increased R-loop formation around this repeat during S phase. We extend this observation by showing that PrimPol suppresses R-loop formation in genes harbouring secondary structure-forming sequences, exemplified by G quadruplex and H-DNA motifs, across the genome in both avian and human cells. Thus, R-loops promote the creation of replication blocks at susceptible structure-forming sequences, while PrimPol-dependent repriming limits the extent of unscheduled R-loop formation at these sequences, mitigating their impact on replication.**

**Keywords** DNA secondary structures; PrimPol; replication; repriming; R-loops
**Subject Categories** DNA Replication, Repair & Recombination
**The EMBO Journal (2019) 38: e99793**

See also: **JEA Reid & T Fischer** (February 2019)

## Introduction

Tracts of repetitive sequence, known as microsatellites or short tandem repeats, occur frequently in vertebrate genomes (Tripathi & Brahmachari, 1991; Clark *et al*, 2006; Willems *et al*, 2014). Many such sequences are capable of forming secondary structures, including hairpins, cruciforms, triplexes (H-DNA) and G-quadruplexes

(G4s), that have the potential to impede DNA replication (Mirkin & Mirkin, 2007). However, the factors that determine whether these sequences pose a barrier to DNA synthesis *in vivo* and the consequences of their doing so are not well understood.

It is well established that long repetitive tracts lead to problems with both replication and transcription. For example, a long tract of polypurine–polypyrimidine (GAA)$_n$ repeats (in which $n$ can exceed 1,500) is linked to the inherited neurodegenerative disorder Friedreich's ataxia (Campuzano *et al*, 1996). These repeats can form H-DNA (Frank-Kamenetskii & Mirkin, 1995), a triplex DNA structure able to block replication both in bacterial, yeast and human cells (Ohshima *et al*, 1998; Krasilnikova & Mirkin, 2004; Chandok *et al*, 2012), which can promote genetic instability of the repeat (Gerhardt *et al*, 2016). Furthermore, these repetitive tracts are prone to form R-loops (Groh *et al*, 2014), three-stranded nucleic acid structures in which nascent RNA hybridises to its complementary DNA template, displacing the non-template DNA strand (Thomas *et al*, 1976). Repetitive sequences can also perturb transcription by reducing RNA polymerase II (RNAPII) elongation (Bidichandani *et al*, 1998; Punga & Buhler, 2010) and lead to deposition of repressive chromatin marks (Saveliev *et al*, 2003; Al-Mahdawi *et al*, 2008). In the case of Friedreich's ataxia, this results in transcriptional silencing of the Frataxin (*FXN*) locus.

Less clear is the impact of the much more common short repetitive tracts found throughout the genome (Clark *et al*, 2006; Willems *et al*, 2014). These have generally not been thought to have any significant impact on replication or transcription. For example, the (GAA)$_n$ repeat in normal alleles of *FXN* ($n < 12$) is not at risk of expansion (Schulz *et al*, 2009), despite the ability of even (GAA)$_9$ to form a stable H-DNA structure at physiological pH *in vitro* (Potaman *et al*, 2004). Further, these "normal" repeats also induce significantly less R-loop formation than disease-length alleles and are not associated with delay of RNAPII or transcriptional silencing (Groh *et al*, 2014). However, it remains unclear whether this apparently inert behaviour is due to these sequences being incapable of forming secondary structures *in vivo* or whether it is the result of activities that counter structure formation and its consequences.

In this paper, we address this question by studying the replication of a short GAA repeat in the *BU-1* locus of chicken DT40 cells.

1  MRC Laboratory of Molecular Biology, Cambridge, UK
2  Sir William Dunn School of Pathology, Oxford, UK
3  Genome Damage & Stability Centre, School of Life Sciences, University of Sussex, Brighton, UK
  *Corresponding author. Tel: +44 1223 267099; E-mail: jes@mrc-lmb.cam.ac.uk

We have previously used this approach to show that G-quadruplexes are able to impede the leading strand polymerase (Sarkies *et al*, 2010, 2012; Schiavone *et al*, 2014; Guilbaud *et al*, 2017) and that repriming, performed by the primase-polymerase PrimPol, is deployed frequently (Schiavone *et al*, 2016). This latter observation suggests that G4s often form impediments during normal replication (Schiavone *et al*, 2016).

Here, we extend this observation to ask what factors drive short tandem repeats to become replication impediments using the poly-purine repeat $(GAA)_{10}$ as a model. We show that this sequence requires PrimPol for its processive replication, demonstrating that these ubiquitous short repeats pose an impediment to DNA synthesis. However, the ability of $(GAA)_{10}$ to impede replication is entirely dependent on DNA:RNA hybrid formation, as overexpression of RNase H1 completely bypasses the requirement for PrimPol. Furthermore, failure of PrimPol-dependent repriming promotes unscheduled R-loop accumulation around the $(GAA)_{10}$ repeat during S phase and, genome-wide, results in higher levels of R-loop formation in genes harbouring secondary structure-forming H-DNA and G4 motifs. These results provide a direct demonstration that R-loop formation can promote DNA sequences with structure-forming potential to become replication impediments. By repriming, PrimPol also prevents the exposure of excessive single-stranded DNA during replication limiting R-loop accumulation in the vicinity of these sequences.

## Results

### Instability of *BU-1* expression monitors replication delay at $(GAA)_n$

We have previously shown that expression instability of the *BU-1* locus in chicken DT40 cells provides a sensitive readout for replication delay at G4 motifs (Schiavone *et al*, 2014). The wild-type locus contains a G4 motif 3.5 kb downstream of the TSS (the +3.5 G4) towards the end of the second intron (Fig 1A), which is responsible for replication-dependent instability of *BU-1* expression under conditions in which G4 replication is impaired (Sarkies *et al*, 2012; Schiavone *et al*, 2014; Guilbaud *et al*, 2017). Failure to maintain processive replication through the +3.5 G4 motif leads to uncoupling of DNA unwinding and DNA synthesis, interrupting normal histone recycling at the fork and the accurate propagation of epigenetic

information carried by post-translational modifications on histone proteins (Fig 1A). This leads to replication-dependent instability of *BU-1* expression manifested as stochastic conversion of the normal "high" expression state to a lower expression level as cells divide (Sarkies *et al*, 2012; Schiavone *et al*, 2014). This expression instability can be readily monitored by flow cytometry analysis of surface Bu-1 protein (Sarkies *et al*, 2012; Fig EV1), providing a simple method to cumulatively "record" episodes of interrupted DNA synthesis at the +3.5 G4.

To model the replication of $(GAA)_n$ repeats, we started with DT40 cells in which the *BU-1* +3.5 G4 motif had been deleted in both alleles (Schiavone *et al*, 2014). $(GAA)_n$ repeats of lengths between $n = 10$ and $n = 75$ were constructed either by synthesis for $n \leq 30$ or, for the longer tracts, using a cloning strategy for highly repetitive sequences (Fig EV2). The repeats were then introduced into the *BU-1A* allele by gene targeting, as previously described (Schiavone *et al*, 2014), to create *BU-1A$^{(GAA)n}$* cells. Following selection cassette removal, cells carrying $(GAA)_{10}$ and $(GAA)_{20}$ in *BU-1A* exhibited wild-type expression levels (Fig 1B). $(GAA)_{30}$ reduced expression of *BU-1A*, while $(GAA)_{50}$ and $(GAA)_{75}$ essentially abrogated expression of the gene (Fig 1B). The reduced expression in cells carrying $(GAA)_{30-75}$ affects the entire population and thus appears to be distinct from the stochastic, replication-dependent loss of expression we have previously reported to be induced by G4 motifs in cells lacking enzymes involved in G4 replication (Sarkies *et al*, 2012; Schiavone *et al*, 2014) or in which G4s are stabilised (Guilbaud *et al*, 2017). Rather, these longer repetitive tracts resulted in the accumulation of chromatin-associated nascent RNA (ChrRNA; Nojima *et al*, 2016) within the locus (Fig EV3A), consistent with impaired expression being due to reduced processivity of RNAPII. As the global reduction of *BU-1* expression in $(GAA)_{30-75}$ alleles precluded the detection of stochastically generated loss variants, we focussed our subsequent analyses on $(GAA)_{10}$ and $(GAA)_{20}$.

Fluctuation analysis for the generation of Bu-1a loss variants confirmed that the presence of $(GAA)_{10}$ at the +3.5 kb position did not affect the stability of *BU-1* expression in a wild-type background (Fig 1C). However, $(GAA)_{20}$ induced modest, but significant, formation of Bu-1a loss variants (Fig 1C), suggesting that this repeat is able to impede replication even in wild-type conditions. We next examined the effect of deleting PrimPol to explore the extent to which repriming mitigates the replication impediment posed by these sequences. The results were striking: the rate at which Bu-1a loss variants were generated in *primpol* cells increased significantly,

**Figure 1. Short (GAA) tracts cause *BU-1* epigenetic instability in *primpol* cells.**

A    Expression instability of the chicken *BU-1* locus as a reporter for replication impediments formed by structure-forming DNA sequences. The leading strand of a replication fork entering the locus from the 3′ end encounters a DNA sequence with structure-forming potential located 3.5 kb downstream of the transcription start site. In wild-type cells, this is a G4 motif, which is replaced by $(GAA)_n$ repeats in this study. Under conditions in which polymerase stalling is prolonged, e.g. loss of G4 processing enzymes or G4 stabilisation (Sarkies *et al*, 2010; Schiavone *et al*, 2014; Guilbaud *et al*, 2017), or repriming is defective (Schiavone *et al*, 2016), the persistence of a putative ssDNA gap leads to a zone of interrupted histone recycling and loss of parental histone modifications. If this loss of modifications involves a control region of the gene, e.g. the promoter, it can result in a change in expression.

B    Flow cytometry for Bu-1a expression in wild-type cells with $(GAA)_n$ tracts of different length knocked into the *BU-1A* locus (in blue). DT40 cells are heterozygous and carry one *BU-1A* and one *BU-1B* allele. All experiments introducing repeats into *BU-1A* are carried out in cells in which the +3.5 G4 has been deleted from both *A* and *B* alleles, to avoid transvection between the alleles (Schiavone *et al*, 2014). Black outline: positive control (wild-type cells); red outline: negative control (cells carrying a genetic disruption of *BU-1*).

C, D    Bu-1a fluctuation analysis of wild-type and *primpol* cells in which the endogenous +3.5 G4 has been deleted (ΔG4) or with $(GAA)_{10}$ and $(GAA)_{20}$ sequence oriented such that it is replicated as the leading (C) or lagging (D) strand template for a fork entering from the 3′ end of the locus as shown in panel (A). At least two independent fluctuation analyses were performed. Circles represent the percentage of Bu-1a loss variants in at least 24 individual clones from these experiments, with mean ± SD reported. ****$P < 0.0001$, *$P \leq 0.05$, ns = not significant; one-way ANOVA.

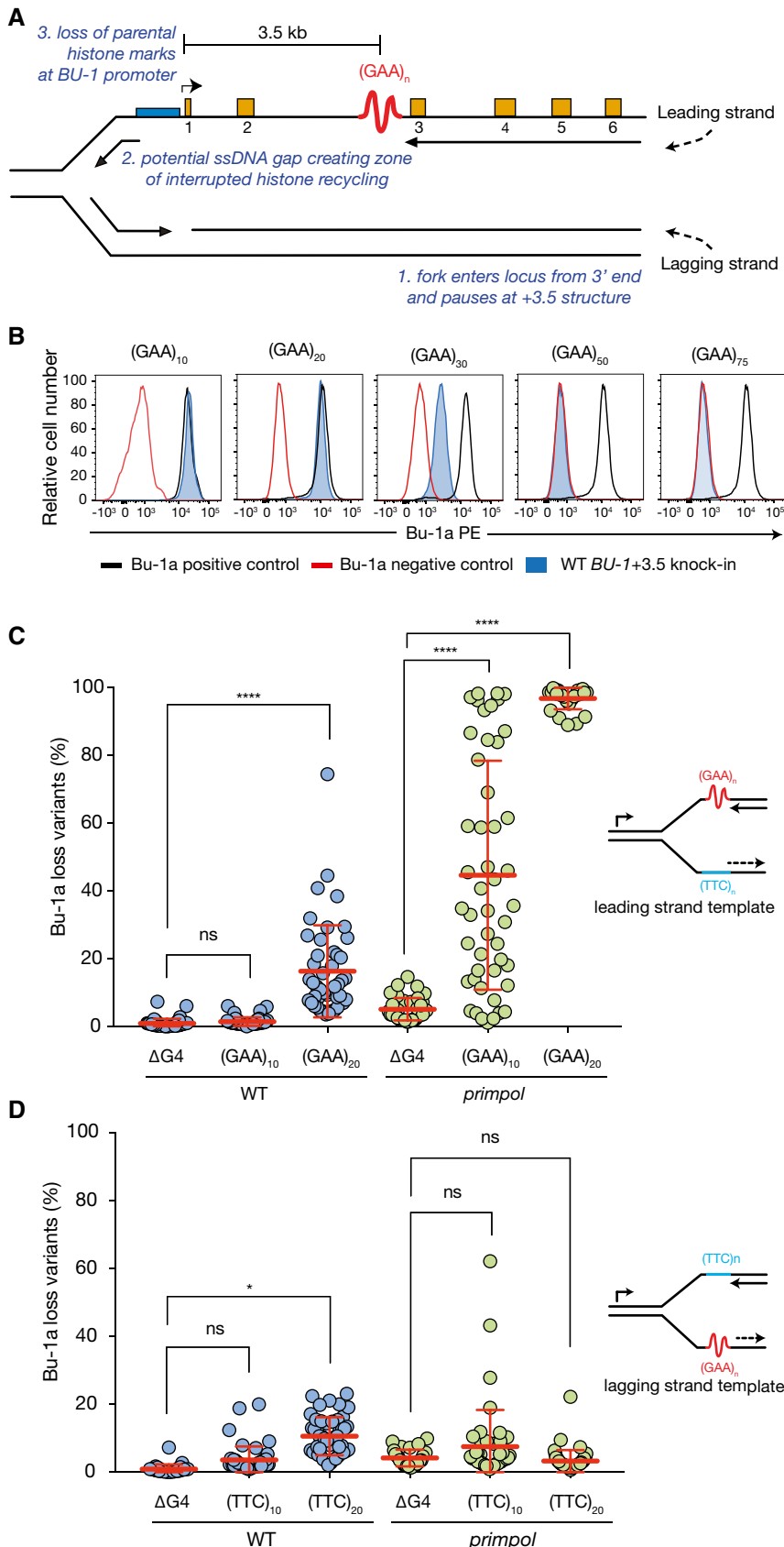

**Figure 1.**

both for $(GAA)_{10}$ and for $(GAA)_{20}$ (Fig 1C). These observations are consistent with repriming preventing significant uncoupling of DNA unwinding and DNA synthesis at these short repeats and thus that they form frequent impediments to otherwise unperturbed DNA replication.

The *BU-1* instability seen in *primpol* cells carrying $(GAA)_{10}$ is comparable to that observed in *primpol* cells harbouring the endogenous +3.5 G4 (Schiavone *et al*, 2016), and similar Bu-1a$^{medium}$ and Bu-1a$^{low}$ expression states, characterised by loss of H3K4me3 and additional DNA methylation respectively, were isolated (Fig EV3B–E). Genetic instability, at a level that could explain the observed formation of Bu-1a loss variants, was not detected (Fig EV3F). The effect of the repeat was orientation dependent (Fig 1D), only producing instability when knocked in such that the purine-rich strand formed the leading strand template for a fork entering from the 3' end of the locus. Together, these observations suggest that the $(GAA)_{10}$ sequence causes epigenetic instability through the same replication-dependent mechanism that we have previously described for G4s.

### The RPA-binding and repriming functions of PrimPol are required to ensure processive replication at the *BU-1* $(GAA)_{10}$ repeat

While PrimPol can perform some translesion synthesis, considerable evidence now supports its main *in vivo* role being repriming (Mouron *et al*, 2013; Keen *et al*, 2014; Kobayashi *et al*, 2016;

Schiavone *et al*, 2016). The repriming function of PrimPol requires the C-terminal zinc finger and RPA-binding motif A (RBM-A), which mediates an interaction with the single-stranded binding protein replication protein A (Wan *et al*, 2013; Guilliam *et al*, 2015, 2017). To confirm that the primary *in vivo* role played by PrimPol in the replication of $(GAA)_{10}$ is indeed repriming, we performed a complementation study by ectopically expressing YFP-tagged human PrimPol in *primpol* cells (Fig 2). Expression of full-length human PrimPol completely restored the stability of *BU-1A* expression in *primpol* cells carrying the $(GAA)_{10}$ repeat. However, neither catalytically inactive PrimPol (D114A, E116A or hPrimPol[AxA]) nor a repriming-defective Zn-finger mutant (C419A, H426A or hPrimPol[ZfKO]) was able to prevent instability of *BU-1* expression (Fig 2). As noted previously, both these constructs confer a growth disadvantage when expressed in DT40 (Schiavone *et al*, 2016), meaning that cells retaining the transgene do not go through as many cell cycles in the course of the experiment resulting in a lower frequency of Bu-1 loss variants (Schiavone *et al*, 2016). Recent work has demonstrated that repriming by PrimPol also requires RPA binding, which is mediated by two RPA-binding motifs, RBM-A and RBM-B. While both RBM-A and RBM-B can bind the same basic cleft in RPA70N *in vitro*, RBM-A appears to play a dominant role *in vivo* (Guilliam *et al*, 2017). Consistent with this observation, expression of hPrimPol[ΔRBM-B] was much more effective at suppressing *BU-1* instability than hPrimPol[ΔRBM-A].

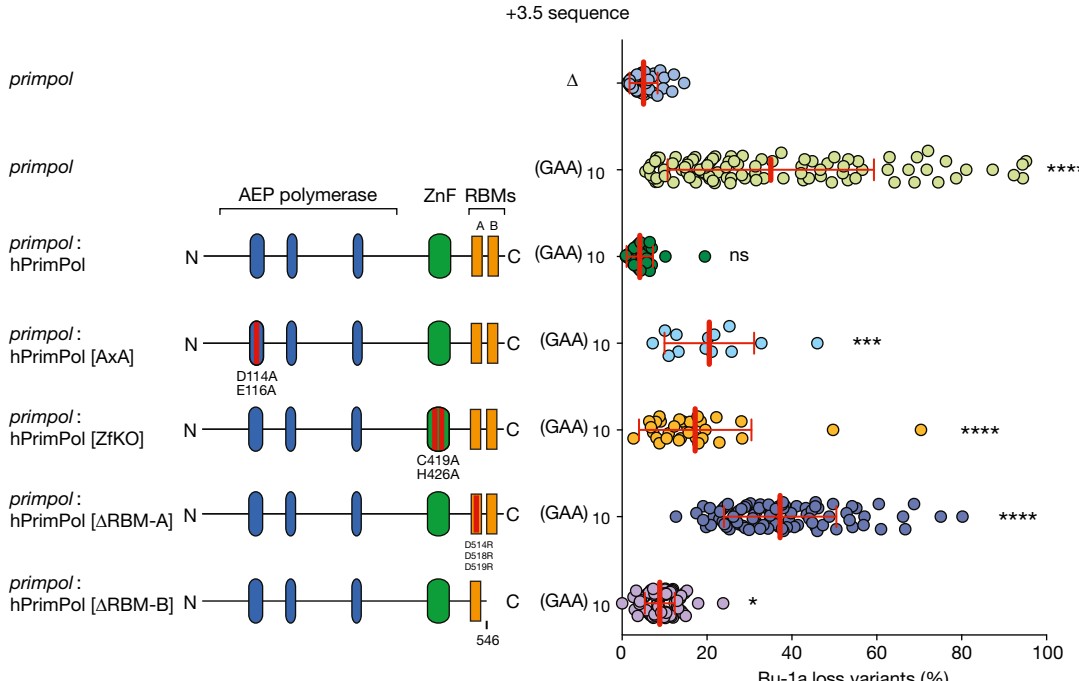

**Figure 2.    The repriming function of PrimPol is required to maintain expression stability of *BU-1* harbouring a $(GAA)_{10}$ repeat.**

Human PrimPol, or mutants, tagged with YFP were expressed in *primpol* cells harbouring $(GAA)_{10}$ sequence in the *BU-1A* locus. Bu-1a- and YFP-double-positive cells were sorted and expanded for 2 weeks, and then analysed for Bu-1a expression variants. For each complementation, at least two independently derived clones were subjected to fluctuation analysis. As previously observed (Schiavone *et al*, 2016), expression of hPrimPol[AxA] and hPrimPol[ZfKO] is deleterious and unstable. Cells expressing these mutations and remaining YFP-positive at the end of the expansion period will have been through fewer divisions than the other lines in this analysis. Pooled results from at least three independent fluctuation analyses are represented with mean ± SD indicated with red bar and whiskers. Statistical significance: ****$P < 0.0001$, ***$P < 0.001$, *$P \leq 0.05$, ns = not significant; Kruskal–Wallis test.

REV1, a Y-family DNA polymerase, is required for maintaining stability of *BU-1* expression when the sequence at the +3.5 kb position is a G4 motif (Sarkies *et al*, 2012; Schiavone *et al*, 2014), which reflects a direct role for REV1 in G4 replication (Sarkies *et al*, 2010; Eddy *et al*, 2014). Replacing the +3.5 G4 in *rev1* cells with $(GAA)_{10}$ repeats did not result in significant destabilisation of *BU-1* expression (Appendix Fig S1), while $(GAA)_{20}$ results in a modest destabilisation of *BU-1* expression, as in wild-type cells. This suggests that the role REV1 plays in maintaining epigenetic stability of *BU-1* is specific to G4 motifs in contrast to PrimPol, the ability of which to reprime is required at both types of secondary structure.

### PrimPol limits R-loop formation around a $(GAA)_{10}$ repeat

The orientation dependence of the GAA tract with respect to *BU-1* instability is in line with the predicted formation of triplex DNA when a polypurine tract is transcribed as the coding strand, but not as the template strand (Grabczyk & Fishman, 1995; Grabczyk *et al*, 2007). *In vitro* studies have shown that the formation of triplexes at $(GAA)_n$ repeats occurs concurrently with the formation of a stable DNA:RNA hybrid between the TTC-rich template strand and the nascent GAA-containing RNA strand (Grabczyk *et al*, 2007). Furthermore, pathological formation of R-loops has been reported at long $(GAA)_n$ repeats $(n \geq 650)$ in immortalised lymphoblasts derived from Friedreich's ataxia patients (Groh *et al*, 2014). These reports, together with the results presented thus far, prompted us to investigate whether R-loops contribute to replication stalling induced by a short GAA tract in *BU-1*.

R-loops can be detected using a DNA:RNA hybrid-specific antibody, S9.6 (Boguslawski *et al*, 1986). We first examined R-loop formation in the $(GAA)_{10}$-containing *BU-1* locus of wild-type cells using DNA:RNA immunoprecipitation (DRIP) followed by quantitative PCR (Fig 3A). In the body of *BU-1* in wild type, the presence of $(GAA)_{10}$ at the +3.5 kb position correlates with a very modest DRIP signal in the vicinity of the repeat. In contrast, *primpol* cells exhibit a highly significant increase in R-loop signal around the repeat. This signal is reduced by treatment of the extracted nucleic acids with RNase H, but not RNase III (Appendix Fig S2), supporting that the detected signal corresponds to RNA:DNA hybrids, rather than dsRNA, which has been reported to cross-react with the S9.6 antibody and to confound R-loop analysis (Phillips *et al*, 2013; Hartono *et al*, 2018). Somewhat surprisingly, the R-loop signal is detected on both sides of the $(GAA)_{10}$ (Fig 3A). While this may, in part, reflect the resolution of the DRIP-qPCR method, it is also consistent with the repeat promoting so-called sticky behaviour, the accumulation of R-loops across the gene body observed in about a quarter of human loci (Sanz *et al*, 2016). Indeed, analysis of RNA DIP-seq data covering *BU-1* in wild-type cells reveals a constitutive coding strand R-loop signal across the locus (Appendix Fig S3). Crucially, the increased gene body R-loop signal in *primpol* cells is abrogated when the +3.5 $(GAA)_{10}$ repeat is deleted (Fig 3A). Together, these data show that PrimPol suppresses R-loop formation associated with this sequence element, rather than playing a more general role in controlling R-loop formation during transcription.

A further striking feature to note in Fig 3A is the strong S9.6 DRIP signal at +11.5 kb, which is in the vicinity of the transcription termination site (Fig 3A). While the presence of this signal is consistent with the previously described formation of R-loops as part of the mechanism of transcription termination in a subset of genes (Skourti-Stathaki *et al*, 2014), it is noteworthy that the signal is increased significantly in the *primpol* mutant. This may be explained by the fact that the region harbours a number of sequences with significant secondary structure-forming potential (Appendix Fig S4).

### R-loop formation is required for $(GAA)_{10}$ to induce expression instability of *BU-1* in PrimPol-deficient cells

We next asked whether formation of a replication block at $(GAA)_{10}$ requires an R-loop. To address this, we overexpressed YFP-tagged chicken RNase H1 carrying a disrupted mitochondrial localisation sequence. RNase H1 degrades R-loops (Stein & Hausen, 1969), and we have previously shown this protein to be stably expressed and active in DT40 cells (Romanello *et al*, 2016). This RNase H1 construct was stably expressed in *primpol BU-1A$^{(GAA)10}$* cells (Appendix Fig S5). This reduced the R-loop signal in the vicinity of the repeat (+3 kb), as did complementation with human PrimPol (Fig 3B). Strikingly, RNase H1 overexpression completely prevented the formation of Bu-1a loss variants in three separate clones of *primpol BU-1A$^{(GAA)10}$*, an effect comparable to removing the $(GAA)_{10}$ repeat itself (Fig 3C). This suggests that DNA:RNA hybrid formation makes a crucial contribution to the ability of $(GAA)_{10}$ to act as a replication impediment and to induce *BU-1* expression instability.

### R-loop stabilisation converts the $(GAA)_{10}$ sequence into a replication impediment

This R-loop dependence of *BU-1* expression instability in *primpol* mutants led us to predict that enforced stabilisation of R-loops might lead the $(GAA)_{10}$ repeat to induce *BU-1* expression instability even in wildtype cells. To achieve this, we overexpressed the 52 amino acid DNA:RNA hybrid binding domain (HBD) of human RNase H1, a fragment previously shown to co-localise with and stabilise DNA:RNA hybrids *in vivo* (Bhatia *et al*, 2014), fused in frame with mCherry separated by a flexible GSGSG linker (Fig 3D). The resulting fusion protein could be stably expressed in DT40 cells as monitored by mCherry fluorescence and Western blotting (Fig 3D and Appendix Fig S6). Expression of the HBD in cells lacking a structure-forming sequence at the +3.5 kb position of *BU-1A* (DT40 *BU-1A$^{\Delta G4}$*) did not induce statistically significant destabilisation of *BU-1* expression compared with the control (Fig 3E). However, when the $(GAA)_{10}$ repeat was present at the +3.5 kb position, we observed significantly greater expression instability. This observation provides further evidence that R-loops are causal in promoting a $(GAA)_{10}$ motif to become a replication block.

### PrimPol curtails R-loop formation during S phase

Since the activity of PrimPol is intimately linked with replication, we hypothesised that specifically removing the R-loops in S phase would suppress $(GAA)_{10}$-induced *BU-1* expression instability. We therefore expressed YFP-tagged chicken RNase H1 fused to a degron sequence from geminin, which ensures protein expression is restricted to S phase (Sakaue-Sawano *et al*, 2008; Fig 4A).

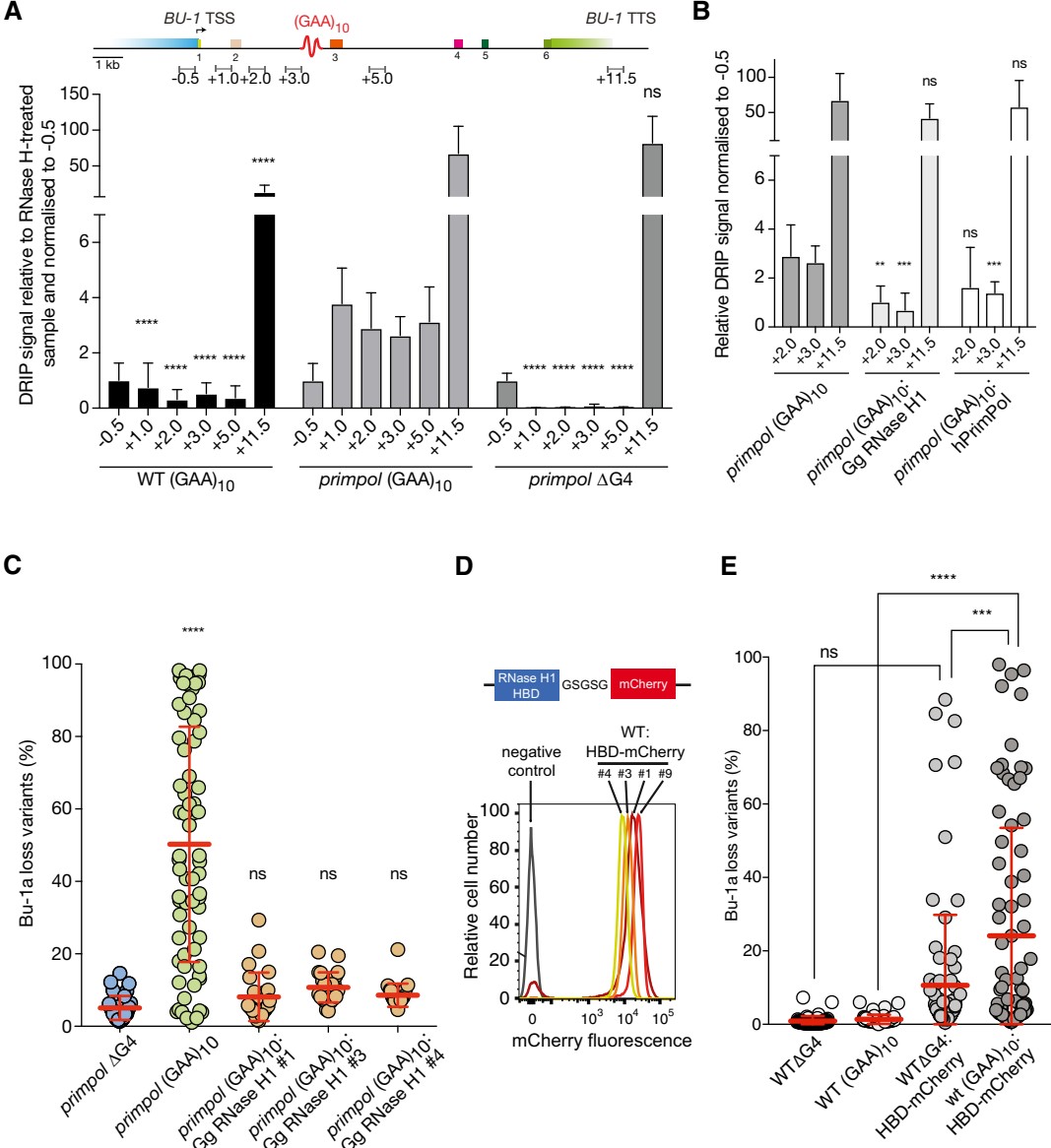

**Figure 3.  R-loops promote (GAA)$_{10}$-dependent epigenetic instability of *BU-1*.**

A   DRIP-qPCR analysis reveals accumulation of R-loops across the *BU-1* locus in *primpol* cells. The DRIP signal was calculated as enrichment over RNase H-treated samples and was normalised to −0.5 kb amplicon. The mean and SD for three biological replicates is presented. An unpaired *t*-test was used to compare differences between matched amplicons in *primpol BU-1A$^{(GAA)10}$* and the other cell lines indicated. ****$P \leq 0.0001$, ns = not significant.

B   DNA:RNA hybrids in *primpol BU-1A$^{(GAA)10}$*:Gg RNase H1 (see also Appendix Fig S5) and *primpol BU-1A$^{(GAA)10}$*:hPrimPol. An unpaired *t*-test on three biological replicates was used to compare differences to *primpol BU-1A$^{(GAA)10}$* for each matched amplicon. The bar represents the mean, and whiskers represent the SD. ***$P \leq 0.001$, **$P \leq 0.01$, ns = not significant.

C   Overexpression of chicken RNase H1 prevents (GAA)$_{10}$-induced *BU-1A* epigenetic instability in *primpol* cells. Fluctuation analysis was performed on three *primpol BU-1A$^{(GAA)10}$* clones. One-way ANOVA was used to calculate the significance of differences in *BU-1* instability between *primpol BU-1A$^{\Delta G4}$* and other cell lines. ****$P \leq 0.0001$, ns = not significant.

D   Diagram of the RNase H1 hybrid binding domain (HBD)–mCherry fusion and flow cytometry expression profiles of the construct in four clones. Western blots of the same four clones are shown in Appendix Fig S6.

E   R-loop stabilisation induces epigenetic instability of *BU-1*. Bu-1a fluctuation analysis of wild-type cells expressing HBD-mCherry. The scatter plots pool results from at least two different clones with matched HBD expression. Mean ± SD reported. ****$P \leq 0.0001$, ***$P \leq 0.001$, ns = not significant; one-way ANOVA.

Expression of this construct was able to prevent instability of *BU-1* expression in *primpol BU-1A$^{(GAA)10}$* (Fig 4B), confirming that R-loops present during S phase are indeed responsible for the (GAA)$_{10}$-dependent destabilisation of *BU-1*.

Next, we asked whether the accumulation of R-loops at the *BU-1* locus of *primpol* cells (Fig 3A) does indeed occur during S phase. Wild-type *BU-1A$^{(GAA)10}$* and *primpol BU-1A$^{(GAA)10}$* cells were synchronised in G1 by double thymidine block and released with

samples taken over the ensuing 6 h (Fig EV4) for monitoring steady state DNA:RNA hybrids by DRIP-qPCR (Fig 4C). This analysis showed the level of DNA:RNA hybrids in the gene body in wild-type cells to be essentially stable through S phase. In contrast, gene body DNA:RNA hybrids increase significantly in *primpol* cells, peaking an hour into S phase. This corresponds to the estimated time that the

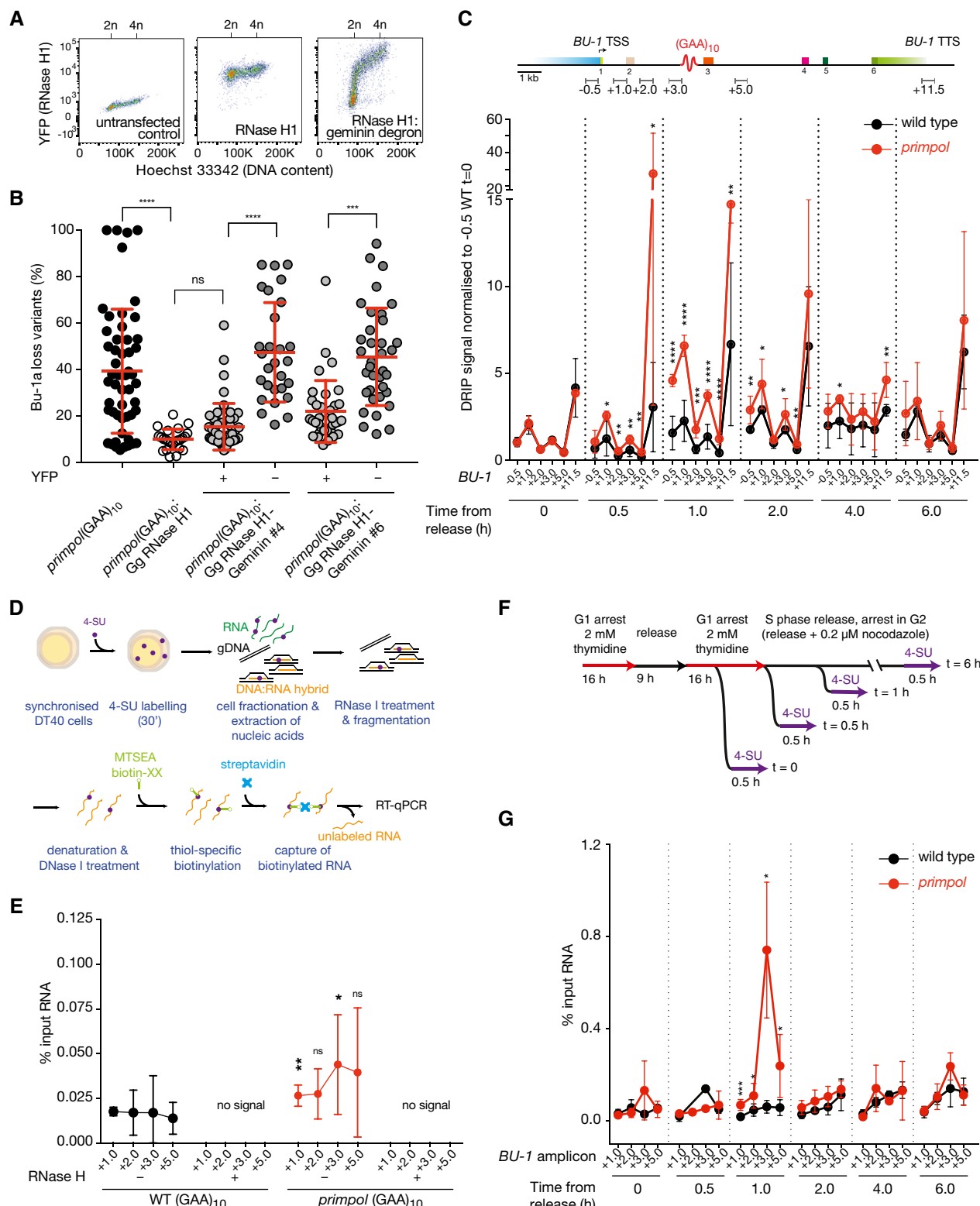

**Figure 4.**

locus is replicated (Schiavone *et al*, 2014). Striking also is the increase in DNA:RNA hybrid signal at the 3′UTR of the *primpol* mutant between 0.5 and 2 h. As noted above, this may be due to a group of structure-forming DNA sequences in the vicinity of the +11.5 kb position. This does not appear to be a general feature of 3′UTRs as a selection of genes that do not contain identifiable structure-forming sequences in their 3′UTR do not exhibit this behaviour (Appendix Fig S7).

The observed increase in the R-loop levels is unlikely to be related to a direct activity of PrimPol on R-loop dissolution as PrimPol lacks the key nucleolytic or helicase activities found in proteins known to directly disrupt R-loops. As the half-life of R-loops in vertebrate cells has been estimated to be around 10 min (Sanz *et al*, 2016), we instead hypothesised that the increase in steady state DNA:RNA hybrids observed in *primpol* cells in S phase was likely due to increased synthesis around the time the locus is replicated. To test this, we used metabolic labelling of nascent RNA with 4-thiouridine (4-SU) to examine active DNA:RNA hybrid formation through S phase (Fig 4D). DNA containing R-loops was gently extracted, and free cytoplasmic and nuclear RNA was degraded with RNase I leaving behind the RNA moiety of R-loops. After heat denaturation and DNase I treatment, surviving 4-SU-labelled RNA was biotinylated and captured with streptavidin beads and its abundance analysed by RT–qPCR (Fig 4D). We observed higher levels of captured nascent DNA:RNA hybrids in the *BU-1* locus of an asynchronous *primpol* culture compared to wild type (Fig 4E), with the increase consistent with higher levels of R-loops detected with S9.6 (Fig 3A). Importantly, treatment with RNase H completely abrogated the signal across the *BU-1* locus. This method to analyse nascent DNA:RNA hybrids also provides support for our central hypothesis free of the potential concerns surrounding the specificity of the S9.6 antibody (Vanoosthuyse, 2018).

Applying this technique to thymidine synchronised cultures (Figs 4F and EV4) revealed a striking spike in R-loop synthesis in the vicinity of the $(GAA)_{10}$ repeat in *primpol* cells an hour after release from G1 arrest (Fig 4G). As noted above, this corresponds closely with the estimated time the locus is replicated (Schiavone *et al*, 2014). Together, these observations are consistent with a model in which excessive R-loop formation during S phase in the absence of PrimPol results from failure to restrict the exposure of

single-stranded DNA gaps produced during interruptions of DNA synthesis.

**PrimPol suppresses R-loop formation in the vicinity of secondary structure-forming sequences throughout the genome**

To explore whether our observations at *BU-1* could be extended to the whole genome, we performed quantitative high-throughput sequencing of S9.6 immunoprecipitated DNA (DRIP-seq) from wild-type and *primpol* DT40 cells. DRIP was performed following treatment of the isolated nucleic acid with RNase A, and the specificity of the DRIP-seq signal was confirmed by pre-treatment of the precipitated nucleic acids with RNase H. To allow quantitation of the DRIP signal, the DT40 samples were spiked with a fixed proportion of *Drosophila* S2 cells to provide an internal control (Orlando *et al*, 2014). Inspection of the distribution of spike-normalised reads, shown across two representative genes in Fig 5A, revealed a strong correlation between wild-type and *primpol* cells, but with higher numbers of reads in peaks in *primpol*. Following peak calling, we examined the global distribution of R-loops across the DT40 genome and found it to be enriched in promoters and terminal regions of genes (Fig 5B), in agreement with previous experiments in human cells (Ginno *et al*, 2013; Skourti-Stathaki *et al*, 2014; Sanz *et al*, 2016).

Peak heights in the wild-type and *primpol* samples were normalised to the mean number of *Drosophila* reads. This revealed a highly significant increase in the height of the DRIP peaks in *primpol* cells (Fig 5C). Between wild type and *primpol*, 84% of peaks were shared suggesting that the loss of PrimPol does not result in the appearance of new peaks, but for any given peak there is a greater DRIP signal, suggesting a higher steady state level of R-loops (Fig 5D). 41% of DRIP peaks overlapped with genes, and 83% of genes with DRIP peaks were shared between wild type and *primpol*. The degree of overlap in the two conditions is not due to the observed change in peak width as the correlation is still observed when allowing 1 kb separation, a much greater distance than the peak width increase. (Appendix Figs S8 and S9). We next asked whether genes with DRIP peaks are enriched for H-DNA motifs. To identify potential H-DNA motifs, we employed the "Triplex" R package (Hon *et al*, 2013), which adopts an approach that allows the identification of

◄

**Figure 4.  Loss of PrimPol leads to unscheduled S phase R-loop formation.**

A   Expression of geminin-tagged chicken RNase H1-YFP. Phases of the cell cycle were determined by staining DNA content in live cells by Hoechst 33342 (*X*-axis). RNase H1-YFP with or without the geminin degron protein is detected on the *Y*-axis. The RNase H1-YFP-geminin degron is degraded in G1. In contrast, RNase H1-YFP levels remain stable irrespective of the phase of the cell cycle. 2n and 4n indicate the chromosome number before and after DNA replication.

B   Bu-1a fluctuation analysis of two independently derived *primpol BU-1A*$^{(GAA)10}$:Gg RNase H1-YFP-geminin degron clones. Since the expression of the RNase H1-YFP-geminin degron construct is not stable (unlike the RNase H1-YFP construct without the degron), Bu-1a expression was assessed separately in the YFP +ve and YFP −ve cells within each clone. Statistical differences calculated the Kruskal–Wallis test. For all panels, at least 36 individual clones were analysed; mean ± SD reported. ****$P \leq 0.0001$, ***$P \leq 0.001$, ns = not significant.

C   DRIP-qPCR for R-loops around the engineered +3.5 $(GAA)_{10}$ repeat in *BU-1* in different phases of the cell cycle. The location of the qPCR amplicons is indicated in the map at the top of the panel. The *BU-1* DRIP signal was normalised to −0.5 kb amplicon in G1-arrested cells (t = 0 h). See Fig EV4 for representative cell cycle synchronisation profiles. Black: wild type; red: *primpol*. Error bars = SD. ****$P \leq 0.0001$, ***$P \leq 0.001$, **$P \leq 0.01$, *$P \leq 0.05$.

D   Workflow for the S9.6-independent detection of newly synthesised R-loops. See Materials and Methods for details.

E   Validation of analysis of nascent DNA:RNA hybrid formation in *BU-1* locus. Enrichment of 4-SU-labelled RNA moiety of DNA:RNA hybrids was calculated relative to input in three independent asynchronous wild-type (black) or *primpol* (red) cells, with or without exogenous RNase H treatment. Error bars = SD. **$P \leq 0.01$, *$P \leq 0.05$, ns = not significant; unpaired *t*-test.

F   Synchronisation and 4-SU pulse labelling scheme to identify nascently formed DNA:RNA hybrids.

G   Newly synthesised R-loops in *BU-1* during S phase in wild type (black) and *primpol* (red). Error bars represent 1 SD of three biological repeats of the experiment. ***$P \leq 0.001$, *$P \leq 0.05$; unpaired *t*-test.

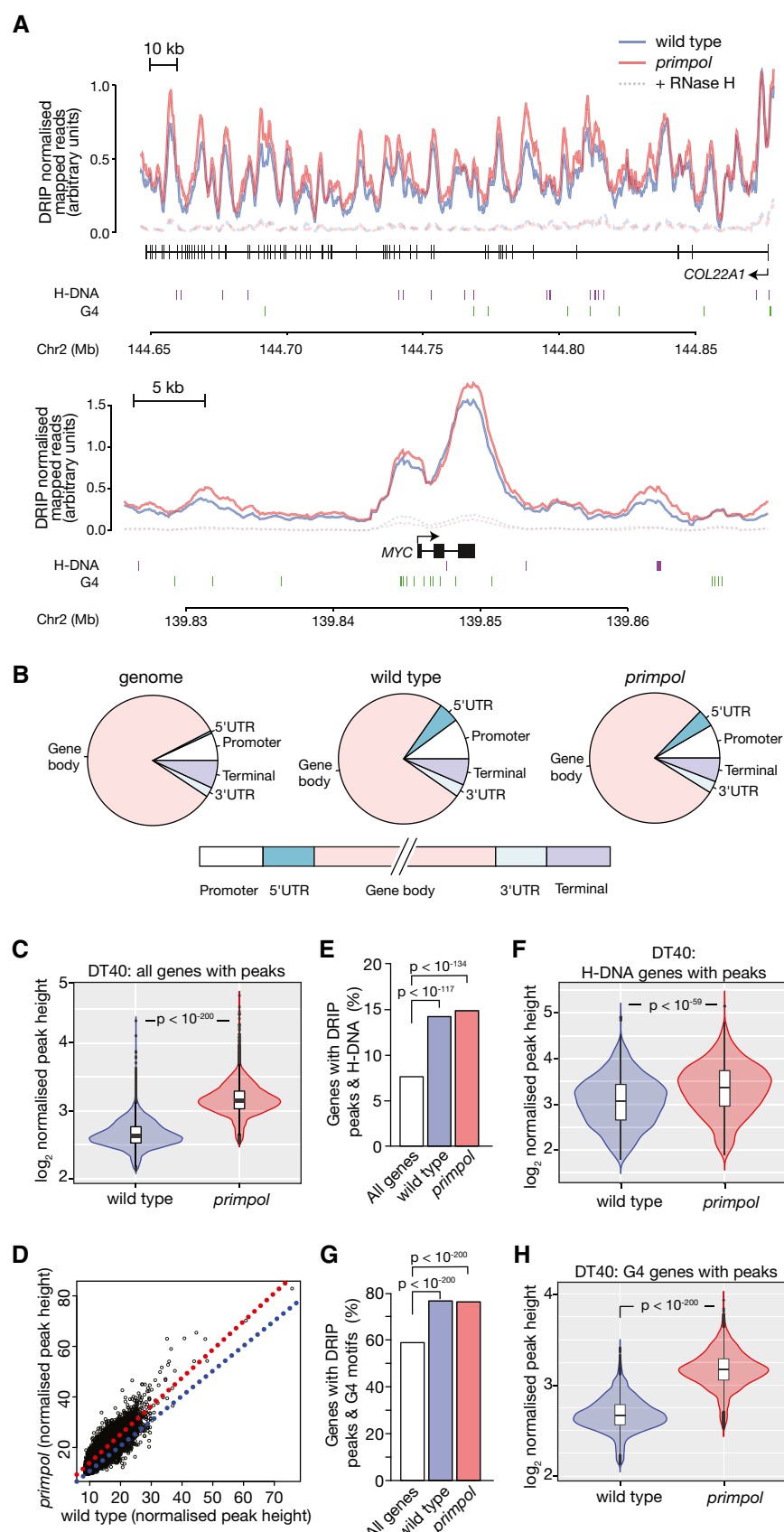

**Figure 5.**

**Figure 5.  PrimPol suppresses R-loop formation in association with DNA secondary structure-forming sequences across the DT40 genome.**

A   Representative normalised DRIP-seq data in two genes *COL22A1*, spanning over 200 kb, and *MYC*. The locations of H-DNA and G4 motifs are shown below the gene map. Wild type in blue; *primpol* in red. The corresponding RNase H-treated samples are dashed. See Materials and Methods for further details of graphic generation.

B   Metagene analysis of DRIP peak distribution in wild-type and *primpol* DT40 cells compared with the distribution of the indicated features in the genome.

C   DRIP peak heights in wild type and *primpol* DT40 normalised to *Drosophila* S2 spike-in. *n* (wild type) = 41,445; *n* (*primpol*) = 48,648.

D   Correlation of normalised DRIP peak heights in the overlapping peaks between wild type and *primpol*. Blue line = 1:1 correlation; red line = linear regression through data.

E   Correlation between H-DNA-forming sequences and all genes (white bar), and genes with DRIP peaks in wild-type (blue) and *primpol* cells (red).

F   Normalised DRIP peak heights in the genes identified as associating with H-DNA.

G   Correlation between G4 motifs ($[G_{3-5}N_{1-7}]_4$) and all genes (white bar), and genes with DRIP peaks in wild-type (blue) and *primpol* cells (red).

H   Normalised DRIP peak heights in the genes identified as associating with G4 motifs ($[G_{3-5}N_{1-7}]_4$).

Data information: *P*-values calculated with Mann–Whitney *U*-test. In violin plots, bar = median; box = interquartile range (IQR); whiskers = upper and lower inner fences (1st/3rd quartile + 1.5*IQR).

sequences with H-DNA-forming potential despite the presence of small imperfections in the sequence as it uses a scoring system based on models of the structures of triplex DNA. The distribution of H-DNAs identified in the chicken genome with this algorithm is shown in Appendix Fig S10. H-DNA sequences were identified as overlapping just under 8% of all genes (see Materials and Methods for further details for determining overlaps). The subset of genes harbouring DRIP peaks was significantly enriched for these sequences, with *c.* 15% of these genes overlapping sequences with H-DNA-forming potential (Fig 5E). Within this set of genes, there was a significant increase in peak height in the *primpol* mutant (Fig 5F). A similar degree of overlap is seen in extragenic DRIP peaks, with 11% of non-genic peaks falling within 1 kb of an H-DNA motif in wild-type cells, and 12% in *primpol*.

Our previous work has shown that G4s are able to induce similar epigenetic instability to the $(GAA)_{10}$ repeat that has been the focus of this study. Further, G4 motifs have been linked to R-loop formation (Duquette *et al*, 2004). We therefore used the regular expression $[G_{3-5}N_{1-7}G_{3-5}N_{1-7}G_{3-5}N_{1-7}G_{3-5}]$ (Huppert & Balasubramanian, 2005) to identify a core subset of G4 motifs in the chicken genome (Appendix Fig S9). G4 motifs identified with this approach overlap with 59% of all genes, while 76% of genes with DRIP peaks overlapped motifs, a significant enrichment (Fig 5G), and again, the heights of the peaks in the *primpol* data set were significantly increased (Fig 5H).

The striking increase in steady state R-loop accumulation in genes containing G4 motifs and our previous demonstration that G4 motifs also potently destabilise *BU-1* expression in *primpol* cells (Schiavone *et al*, 2016) prompted us to ask whether R-loops

also promote G4 motifs to become replication impediments. Forcing RNase H1 expression resulted in a highly significant reduction in *BU-1* instability induced by the natural +3.5 G4 in four separate RNase H1-expressing clones (Fig EV5), demonstrating that R-loop formation increases the probability of this G4 forming a significant replication impediment, but that it is not essential for it to do so.

Finally, we asked whether loss of PrimPol also affected R-loop levels in human cells. PrimPol was disrupted using CRISPR/Cas9 editing in the induced pluripotent stem cell line BOBSC (Yusa *et al*, 2011). Genome-wide R-loops were isolated with S9.6 immunoprecipitation followed by a modified sequencing protocol, RNA DIP-seq, that monitors the RNA moiety within the DNA:RNA hybrids. An example of the signal, normalised to a spike-in control of DT40 cells, across the *SKI* locus in wild type and *primpol* BOBSC (Fig 6A) shows the same pattern of R-loop enrichment observed in a previous study (Sanz *et al*, 2016). It also demonstrates the overlap of sites of R-loop formation between wild type and *primpol* and the increase in peak heights in *primpol*. Following peak calling, we determined that the overall distribution of R-loops across a metagene is similar to that in DT40 (Fig 5B) and that described previously for human cells, with an enrichment at promoter and terminus regions of genes (Fig 6B; Sanz *et al*, 2016). The increase in the height of existing peaks in *primpol* cells, evident in the *SKI* locus (Fig 6A), is confirmed by genome-wide analysis. 66% of wild-type and *primpol* peaks overlap with a highly significant increase in peak height in the *primpol* mutant (Fig 6C and D). Again, this correlation is independent of the observed increase in peak width (Appendix Fig S8). As in DT40,

**Figure 6.  PrimPol suppresses R-loop formation in association with DNA secondary structure-forming sequences in BOBSC iPS cells.**

A   Representative normalised RNA DIP-seq data in the *SKI* locus. Wild type in blue; *primpol* in red. The locations of H-DNA and G4 motifs are shown below the gene map. The corresponding RNase H-treated samples are dashed. Since so little material was recovered following RNase H treatment, all samples were pooled prior to library generation.

B   Metagene analysis of RNA-DIP peak distribution in wild-type and *primpol* BOBSC cells compared with the distribution of the indicated features in the genome.

C   RNA DIP-seq peak heights in wild-type and *primpol* BOBSC cells normalised to a DT40 spike-in. *n* (wild type) = 32,740; *n* (*primpol*) = 33,721.

D   Correlation of normalised RNA DIP-seq peak heights in the overlapping peaks between wild type and *primpol*. Blue line = 1:1 correlation; red line = linear regression through data.

E   Correlation between H-DNA-forming sequences and all genes (white bar), and genes with DRIP peaks in wild-type (blue) and *primpol* cells (red).

F   Correlation between G4 motifs ($[G_{3-5}N_{1-7}]_4$) and all genes (white bar), and genes with DRIP peaks in wild-type (blue) and *primpol* cells (red).

G   Normalised RNA DIP-seq peak heights in the genes identified as associating with H-DNA.

H   Normalised RNA DIP-seq peak heights in the genes identified as associating with G4 motifs ($[G_{3-5}N_{1-7}]_4$).

Data information: *P*-values calculated with Mann–Whitney *U*-test. In violin plots, bar = median; box = interquartile range (IQR); whiskers = upper and lower inner fences (1st/3rd quartile + 1.5*IQR).

the genes containing peaks were associated with H-DNA and G4 motifs (Fig 6E and F), the length distribution of which is comparable between the two species (Appendix Fig S10). In both cases, the mean peak height was significantly higher in *primpol* cells (Figs 6G and H), demonstrating that loss of PrimPol also results in increased R-loop formation in the vicinity of DNA secondary structures in human cells.

# Discussion

### A requirement for PrimPol reveals that (GAA)$_{10}$ forms a replication impediment

The (GAA)$_{10}$ repeat upon which this study has focussed is typical of widespread short tandem repeats found throughout vertebrate

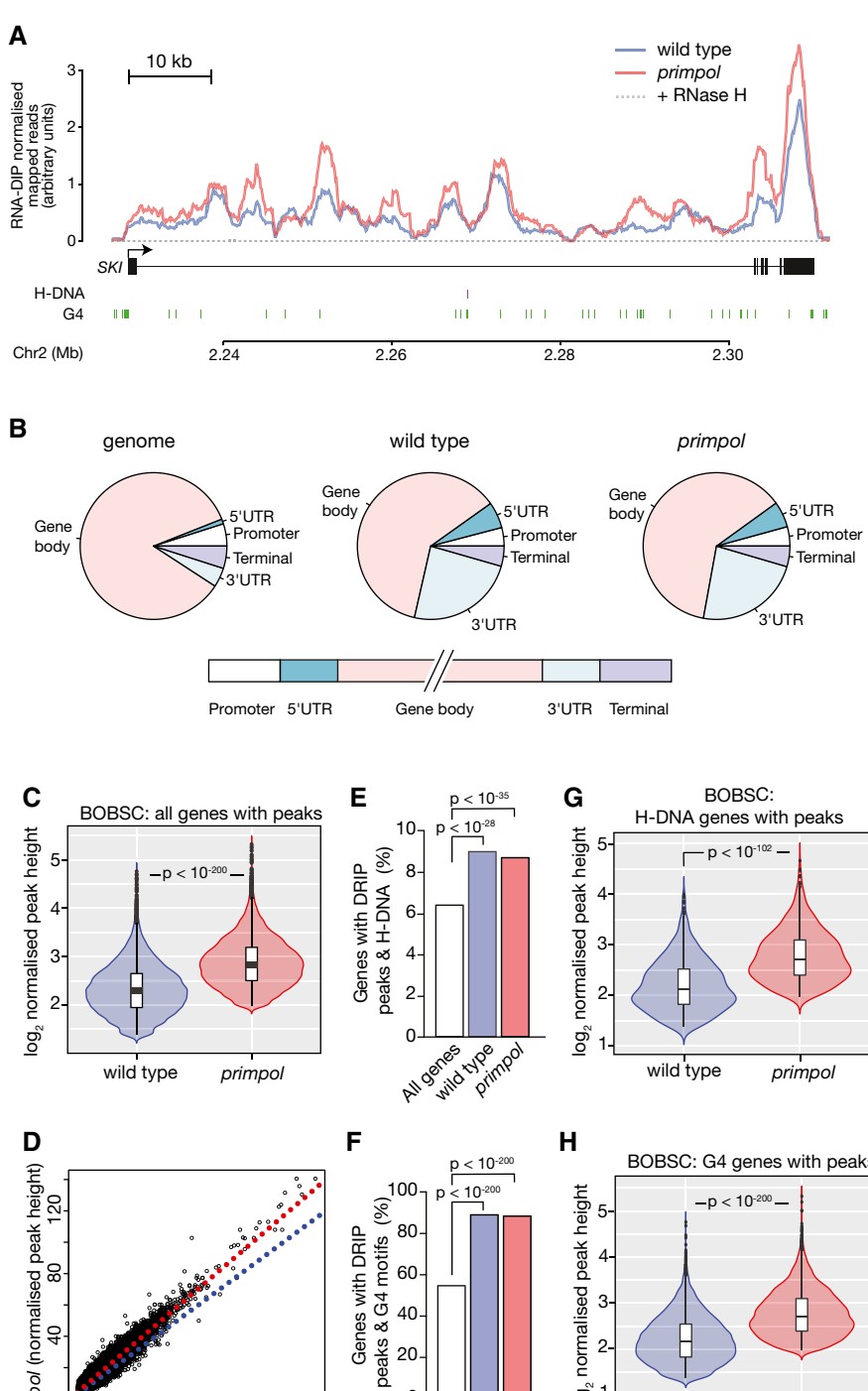

**Figure 6.**

genomes (Willems *et al*, 2014). Repeats of this length have not been previously linked to detectable disturbances in replication or transcription *in vitro* (Bidichandani *et al*, 1998; Ohshima *et al*, 1998) despite their potential to form triplex structures at physiological pH (Potaman *et al*, 2004). Our previous work supports a model in which instability of *BU-1* expression induced by G4s results from uncoupling of DNA unwinding from leading strand DNA synthesis (Sarkies *et al*, 2010; Schiavone *et al*, 2014; Šviković & Sale, 2017). This uncoupling can extend up to *c.* 4.5 kb (Schiavone *et al*, 2014), consistent with earlier observations in both mammalian and yeast cells (Lehmann, 1972; Lopes *et al*, 2006), and is mitigated by PrimPol-dependent repriming (Schiavone *et al*, 2016). We now show that this repriming activity is also frequently deployed at a model short tandem repeat, of a form found commonly in vertebrate

genomes, demonstrating that these sequences can indeed form replication impediments.

### The nature of the replication impediment formed by $(GAA)_{10}$

$(GAA)_n$ repeats, in common with other polypurine–polypyrimidine tracts, are capable of forming triplex secondary structures in which a third strand anneals through Hoogsteen base pairing. This tendency has been linked to the detrimental effect of long GAA repeats on transcription elongation (Bidichandani *et al*, 1998; Punga & Buhler, 2010) through the trapping of transcribing RNA polymerase II (Grabczyk & Fishman, 1995). However, the very act of transcription also promotes formation of secondary structures (Lilley, 1980; Kouzine *et al*, 2017), including triplexes (Grabczyk &

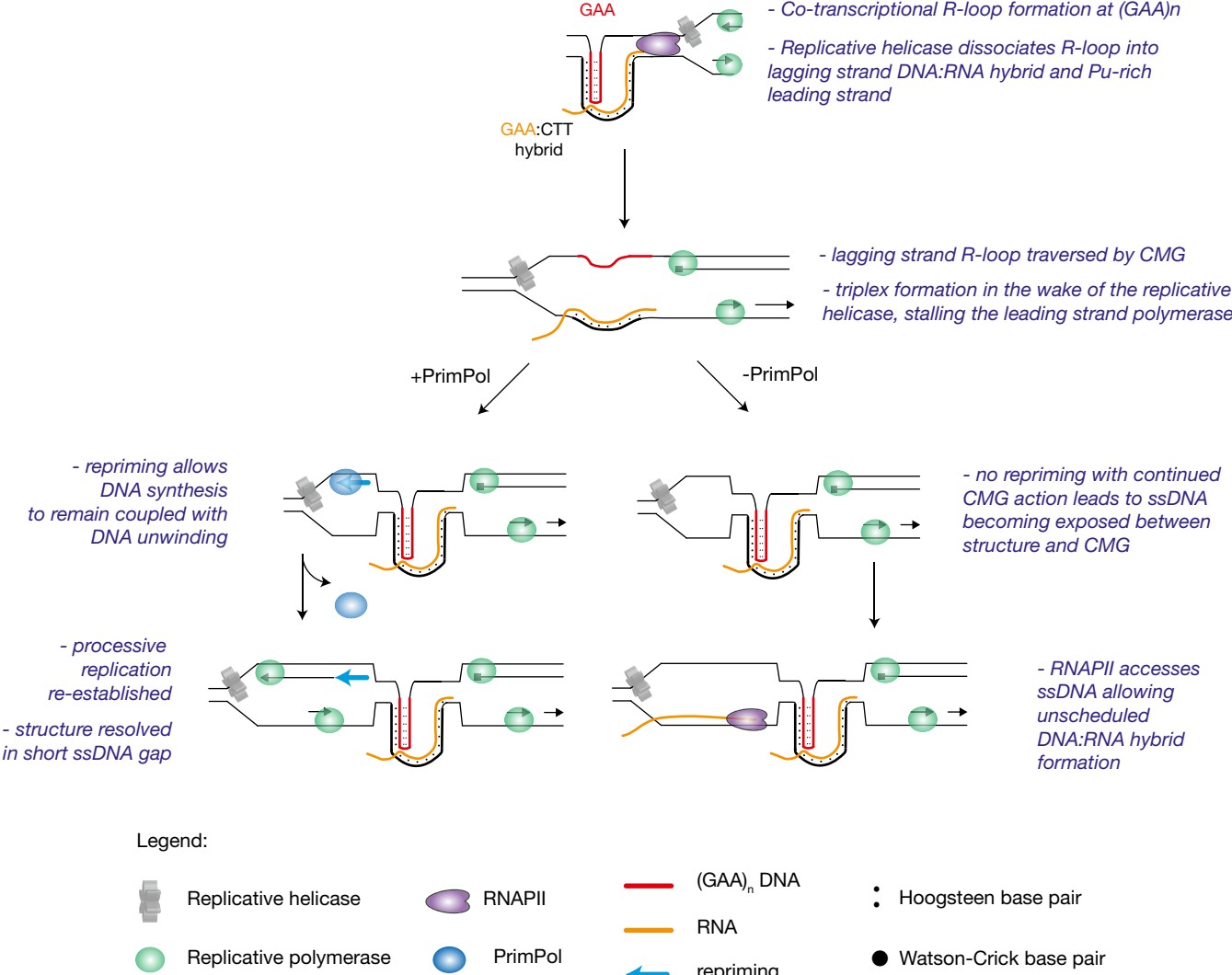

**Figure 7. A model for how PrimPol suppresses S phase R-loops in the vicinity of DNA secondary structures.**
During transcription, an R-loop is formed at the GAA tract, which can promote triplex or H-loop formation (Neil *et al*, 2018). We propose this structure is dissociated by replicative helicase into the purine-rich leading strand and DNA:RNA hybrid on the lagging strand. In the wake of the replicative helicase, the triplex/H-loop may reform between DNA:RNA hybrid and purine-rich sequence, blocking the leading strand polymerase. In wild-type cells, PrimPol is recruited to the single-stranded gap, where repriming allows DNA synthesis to remain coupled to fork progression. In the absence of PrimPol, however, the activity of replicative helicase exposes long stretches of ssDNA. RNA polymerase II gains access to such template, leading to unscheduled DNA:RNA hybrid formation.

Fishman, 1995; Kouzine *et al*, 2004). This is likely driven by structure formation releasing the negative supercoiling generated in the wake of translocating RNA polymerase (Liu & Wang, 1987; Levens *et al*, 2016). Similar topology-induced structure formation could also contribute to leading strand secondary structure formation behind the replicative helicase (reviewed in Kurth *et al*, 2013; Yu & Droge, 2014).

We show that $(GAA)_{10}$ requires an RNase H1-sensitive R-loop in order to create a replication impediment that requires PrimPol-dependent repriming. It is well established that formation of DNA:RNA hybrids coincides with the sequences able to adopt non-B DNA structures (Duquette *et al*, 2004; Grabczyk *et al*, 2007) and is favoured in a negatively supercoiled DNA template (Roy *et al*, 2010). R-loops have been implicated as a major factor in the severity of head-on collisions between the replication and the transcriptional machinery (Hamperl *et al*, 2017). However, a direct head-on collision with transcribing RNA polymerase is likely to halt the entire replisome (Pomerantz & O'Donnell, 2010), precluding the displacement of parental nucleosomes caused by the uncoupling between the replicative helicase and DNA synthesis. It is difficult to reconcile this type of stall with the involvement of PrimPol. Specifically, the DNA- and RPA-binding activities of the enzyme suggest the transient formation of ssDNA, which most likely arises as a result of uncoupling of replicative helicase from the replicative polymerases, and which is the basis for *BU-1* expression instability.

How then can a $(GAA)_n$ repeat generate the uncoupling of DNA unwinding and leading strand DNA synthesis necessary to induce expression instability of *BU-1*? We propose that transcription of the $(GAA)_{10}$ repeat generates an R-loop (Fig 7). During replication, the approaching replicative helicase traverses the transcription complex by displacing the RNA polymerase (Pomerantz & O'Donnell, 2010) or by reorganising the helicase itself (Huang *et al*, 2013; Vijayraghavan *et al*, 2016). Biophysical calculations show that DNA:RNA hybrids are sufficiently thermodynamically stable to survive the accumulation of positive supercoiling generated ahead of the replicative helicase (Belotserkovskii *et al*, 2013). Since the eukaryotic replicative helicase tracks on the leading strand (Douglas *et al*, 2018), we suggest that the DNA:RNA hybrid could remain intact on the lagging strand during passage of the helicase. Behind the replicative helicase, the persistent lagging strand DNA:RNA hybrid may re-trap the purine-rich leading strand through triplex formation. The resulting R:R•Y hybrid triplex, recently termed an H-loop (Neil *et al*, 2018), could then block leading strand synthesis (Samadashwily & Mirkin, 1994). This model is consistent with the observation that the depletion of DNA:RNA hybrids through overexpression of RNase H1 completely abolishes $(GAA)_{10}$-dependent *BU-1* expression instability in PrimPol-deficient cells. An alternative explanation for the creation of a leading strand impediment is the formation of a DNA triplex stabilised by an adjacent DNA:RNA hybrid, of the form proposed by Grabczyk and Fishman (1995). In either event, continued helicase activity would result in exposure of ssDNA ahead of the stalled replicative polymerase, which through being bound by RPA promotes the recruitment of PrimPol. Repriming close to the structure then allows DNA synthesis to remain coupled to unwinding leaving the triplex in a small gap to be disassembled post-replicatively (Fig 7). Whether loss of PrimPol completely disables leading

strand repriming or whether PRIM1 (the primase associated with Pol α) can substitute to some extent remains unclear. Recent work with a reconstituted yeast replisome suggests that leading strand repriming by PRIM1 is intrinsically inefficient (Taylor & Yeeles, 2018), suggesting that the ssDNA generated by the helicase–polymerase uncoupling event will ultimately be replicated either by restart of the stalled fork or by a fork arriving from the opposite direction.

### Loss of PrimPol-mediated repriming at structured DNA promotes S phase R-loop accumulation

In the absence of PrimPol, continued unwinding of the parental duplex by the replicative helicase in the context of a continued stalling of DNA synthesis would create a more extensive region of ssDNA. That this results in increased R-loop formation in *primpol* cells during the time the locus is replicated implies that RNAPII continues to transcribe despite its template remaining single-stranded. This idea is consistent with both biochemical (Kadesch & Chamberlin, 1982) and *in vivo* reports (Ohle *et al*, 2016; Michelini *et al*, 2017) demonstrating that DNA:RNA hybrid formation can occur at ssDNA generated by resection of DNA ends at double-stranded DNA breaks. We suggest that ssDNA generated as a consequence of helicase–polymerase uncoupling, unmitigated by repriming, could also act as a substrate for unscheduled RNAPII transcription and DNA:RNA hybrid formation (Fig 7).

The results we present here establish two important mechanistic points concerning the relationship between R-loops and impeded replication. First, we show that R-loops are able to promote short sequences with structure-forming potential to become replication impediments, requiring the repriming activity of PrimPol to maintain their processive replication. Second, failure to reprime at these sequences increases R-loop formation. We suggest this is due to exposure of excessive single-stranded DNA during S phase, potentially increasing unscheduled access of RNAPII. An increase in R-loops associated with structure-forming DNA sequences is seen throughout the genome in PrimPol-deficient cells, but principally in regions in which R-loops are already formed in wild-type cells, particularly in transcribed regions. This suggests that repriming plays a particularly important role in allowing cells to manage the complex challenges created by clashes between transcription and replication.

## Materials and Methods

### Cell culture and transfection

DT40 cell culture, cell survival assays, fluctuation analysis for generation of Bu-1 loss variants (Fig EV1), and genetic manipulation of the *BU-1* locus were performed as previously described (Simpson & Sale, 2003, 2006; Schiavone *et al*, 2014). *Drosophila* S2 cells were grown in Insect-XPRESS Protein-Free Insect Cell Medium with L-glutamine (Lonza), supplemented with 1% penicillin–streptomycin at 27°C, ambient $CO_2$ with 105 rpm agitation. BOBSC human induced pluripotent stem (hiPS) cells (Yusa *et al*, 2011) were cultured feeder-free on dishes coated with Vitronectin XF (07180;

Stem Cell Technologies) in Essential 8 Flex media (A2858501; Thermo Fisher Scientific) at 37°C and 5% $CO_2$. Cells were split 1:10–1:15 every 3–4 days depending on confluence. All cells tested negative for mycoplasma.

### CRISPR/Cas9-mediated gene disruption in human cells

Guide RNA sequences used for disrupting *PRIMPOL* in BOBSC iPS cell lines are listed in the Appendix. Each gRNA sequence was cloned into pX458 (Ran *et al*, 2013). A targeting construct carrying puromycin selection marker was constructed by Gibson assembly using PCR-amplified 5′ and 3′ homology arms (see Appendix Table for all oligonucleotides used in this study). Equimolar amounts of targeting construct, gRNA expression vectors and Cas9 expression vectors were delivered by Amaxa electroporation. Puromycin-resistant clones were genotyped by PCR and Sanger sequencing (Shen *et al*, 2014).

### Molecular cloning and transgene constructs

For transgene expression, cDNA was cloned in frame with fluorescent protein (mCherry or YFP) in the polylinker of pXPSN2 (Ross *et al*, 2005). The expression module was released with SpeI digestion and subcloned into pBluescript-based vectors containing a loxP flanked puromycin or blasticidin S selection cassettes (Arakawa *et al*, 2001), which were transfected into DT40 via electroporation. Primers used for molecular cloning are listed in the Appendix. The hPrimPol-YFP, hPrimPol [AxA]-YFP and hPrimPol[ZfKO]-YFP constructs for complementation of *primpol* DT40 were previously described (Schiavone *et al*, 2016). cDNAs for PrimPol RPA binding mutants (ΔRBM-A and ΔRBM-B) were PCR-amplified from previously described vectors (Guilliam *et al*, 2017) with primers listed in the Appendix. Similarly, chicken RNase H1 (lacking the mitochondrial localisation sequence) was PCR-amplified from DT40 cDNA and cloned in frame with YFP on the C-terminus. To produce a cell cycle-regulatable RNase H1, a fragment of human geminin corresponding to amino acids 1–110 was PCR-amplified from pLL3.7m-Clover-Geminin(1-110)-IRES-mKO2-Cdt(30-120) from the Fucci4 system (Bajar *et al*, 2016) and cloned in frame to the C-terminus of chicken RNase H1 ΔMLS-YFP. The hybrid binding domain was amplified from human cDNA using primers previously published (Bhatia *et al*, 2014) and fused to C-terminal mCherry via flexible GSGSG linker. Uninterrupted GAA tracts were created using a previously developed strategy (Scior *et al*, 2011); see Fig EV2 for details. The acceptor plasmid was created by inserting a linker containing a $(GAA)_{20}$ repetitive tract flanked by recognition sites for the restriction enzymes BbsI, BsmBI and NcoI into pBluescript SK(+). Plasmids containing the repeats were transformed into a strain of *Escherichia coli* lacking the SbcCD nuclease, DL733 (Connelly *et al*, 1997), a kind gift from David Leach, to avoid excision of secondary structures caused by the repeat. The confirmed uninterrupted repeats with minimal flanking sequence were released by MluI digest, subcloned into the BU-1 targeting construct and screened for orientation by Sanger sequencing.

### Chromatin immunoprecipitation

Chromatin immunoprecipitation (ChIP) was performed as previously described (Schiavone *et al*, 2014). Solubilised chromatin was

diluted and immunoprecipitated overnight using the following antibodies: histone H3 (ab1791; Abcam), H3K4me3 (9727; Cell Signaling), H3K36me3 (ab9050; Abcam) and normal rabbit IgG (2729; Cell Signaling). Primer sequences are provided in the Appendix. Enrichment was normalised to H3 and reported relative to wild type.

### Bisulphite sequencing

Bisulphite conversion of genomic DNA was performed using EZ DNA Methylation-Gold Kit (Zymo Research) as per the manufacturer's instructions. PCR was performed with ZymoTaq (Zymo Research) and primers compatible with bisulphite-converted DNA (Appendix) for 40 cycles according to the manufacturer's instructions. PCR products were purified, digested with NotI and SacI, size-selected by gel extraction and cloned into pBK CMV. Plasmids from individual bacterial colonies were sequenced by GATC Biotech.

### Chromatin-associated RNA (ChrRNA) extraction

RNA associated with the chromatin was extracted as described previously (Nojima *et al*, 2016). Briefly, DT40 cells were lysed in HLB + N [10 mM Tris–HCl (pH 7.5), 10 mM NaCl, 2.5 mM $MgCl_2$ and 0.5% (vol/vol) NP-40] and passed through a 10% sucrose cushion. Nuclei were then resuspended in 125 μl NUN1 buffer [20 mM Tris–HCl (pH 7.9), 75 mM NaCl, 0.5 mM EDTA and 50% (vol/vol) glycerol], mixed with 1.2 ml NUN2 [20 mM HEPES–KOH (pH 7.6), 300 mM NaCl, 0.2 mM EDTA, 7.5 mM $MgCl_2$, 1% (vol/vol) NP-40 and 1 M urea], vortexed vigorously and spun for 10 min at 16,000 *g*. Chromatin pellets were digested twice with DNase I (NEB, M0303) and Proteinase K. ChrRNA was extracted with QIAzol (QIAGEN) and converted to cDNA using QuantiTect Reverse Transcription Kit (QIAGEN) as recommended by the manufacturer. The enrichment of RNA across the *BU-1* locus was analysed by qPCR and the signal normalised to GAPDH.

### DNA:RNA immunoprecipitation (DRIP)

Extraction of R-loops is largely based on methods previously described (Groh *et al*, 2014). Ten to 30 million DT40 cells were harvested by centrifugation, washed in 25 ml cold PBS and lysed in cell lysis buffer (85 mM KCl, 5 mM PIPES (pH 8.0), 0.5% NP-40) for 10 min on ice. Nuclei were gently pelleted (1,000 *g*, 10 min), equilibrated in nuclei lysis buffer (50 mM Tris–HCl (pH 8.0), 1.2 mM EDTA, 1% SDS) and then incubated overnight at 37°C upon addition of Proteinase K (Thermo Fisher). SDS and contaminating proteins were removed by adding 5 M KOAc (pH 5.5) and centrifuging at high speed for 15 min. DNA was precipitated from the supernatant with glycogen (Santa Cruz Biotechnology) and isopropanol. DNA was pelleted, gently washed several times with 70% EtOH and rehydrated in 10 mM Tris–HCl (pH 8.0).

Genomic DNA containing DNA:RNA hybrids was digested overnight with a restriction enzyme cocktail containing BamHI, NcoI, PvuII, ApaLI and NheI, yielding an average fragment size of 1 kb. Samples were subsequently diluted to 5 ml with IP dilution buffer [16.7 mM Tris–HCl (pH 8.0), 1.2 mM EDTA, 167 mM NaCl, 1.1% Triton X-100, 0.01% SDS], pre-cleared for 2 h with 30 μl Protein G Sepharose beads (Dharmacon) and immunoprecipitated with 10 μg S9.6 antibody overnight at 4°C. Subsequent steps are essentially the

same as for ChIP. Briefly, captured immunocomplexes were washed with low-salt [0.1% SDS, 1% Triton X-100, 20 mM Tris–HCl (pH 7.5), 165 mM NaCl, 2 mM EDTA], high-salt [0.1% SDS, 1% Triton X-100, 20 mM Tris–HCl (pH 7.5), 500 mM NaCl, 2 mM EDTA] and LiCl [1% NP-40, 1% deoxycholate, 10 mM Tris–HCl (pH 7.5), 250 mM LiCl, 1 mM EDTA] wash buffers and TE buffer [10 mM Tris–HCl (pH 7.4), 1 mM EDTA]. DNA:RNA hybrids were eluted for 2 h at 65°C in elution buffer (1% SDS, 100 mM NaHCO$_3$) and purified with PCR Purification Kit (QIAGEN). The specificity of the pull-down was tested with RNase H and RNase III treatments prior to immunoprecipitation: one-third of the digested material was treated with 25 U of RNase H (NEB, M0297), or with 10 U of RNase III (Ambion, AM2290) in appropriate buffers overnight at 37°C, with the subsequent steps performed as described above. The signal across *BU-1* locus was normalised to RNase H background signal and baselined to 28S rDNA.

## Cell cycle synchronisation and 2D cell cycle analysis

G1 phase synchronisation of DT40 cells was achieved by double thymidine block. Cells were treated overnight with 2 mM thymidine, released for 9 h and again treated with thymidine overnight, after which cells were released into medium containing 0.2 μM nocodazole to prevent cells entering mitosis. Upon release from thymidine block, cells in different cell cycle phases were harvested to be analysed or pulse-labelled with BrdU. Five to 10 million DT40 cells were pulse-labelled with 50 μM BrdU for 30 min in complete medium at 37°C. BrdU staining was performed as previously described (Frey *et al*, 2014).

## Capture of 4-SU-labelled nascent DNA:RNA hybrids

To label nascently formed DNA:RNA hybrids, 150–250 million DT40 cells were resuspended in 10 ml of warm complete medium supplemented with 100 μM 4-thiouridine (4-SU; Sigma-Aldrich) and incubated for 30 min at 37°C. 4-SU incorporation was terminated by adding ice-cold PBS, after which nuclei were extracted as described for ChrRNA extraction. The nuclear pellet was divided, and equivalent of 20–50 million nuclei were lysed in 700 μl of nuclear lysis buffer (25 mM Tris–HCl (pH 7.4), 1% SDS, 5 mM EDTA and 0.125 mg/ml Proteinase K) overnight at 37°C with agitation. SDS and digested proteins were removed with 1 M potassium acetate (pH 5.5) and nucleic acids precipitated with isopropanol. Any soluble ssRNA was degraded by treating the nucleic acids with 25 U of RNase I (Ambion) in TNE buffer (10 mM Tris–HCl (pH 7.4), 100 mM NaCl, 1 mM EDTA) for 30 min at 37°C; to remove RNase I (Ambion, AM2294), 5 μg of Proteinase K was added and reaction incubated for a further 2 h, after which the nucleic acids were purified with phenol:chloroform:isoamyl alcohol (25:24:1) and precipitated with 0.3 M NaCl. Nucleic acids were resuspended in water and fragmented using Bioruptor Plus (Diagenode) to average size of 500 bp (30 cycles, 30″ ON, 30″ OFF at high output). Appropriate specificity controls were performed at this point identically to the DRIP protocol.

Sheared nucleic acids were supplemented with 1× DNase I Reaction Buffer (New England Biolabs), denatured for 5 min at 95°C and snap-cooled on ice to release RNA moiety of DNA:RNA hybrids. To digest the DNA component, denatured nucleic acids were incubated

with 15 U of RNase-free DNase I (NEB) at 37°C. RNA moiety of DNA:RNA hybrids was extracted with QIAzol (QIAGEN) as per the manufacturer's instructions and precipitated with glycogen at −80°C. RNA was collected by centrifugation, washed with 70% ethanol and resuspended in 217 μl of RNase-free water. 2% of recovered RNA was reserved as total input for normalisation. 4-SU-containing RNA was further labelled with thiol-specific and reversible biotinylation reagent MTSEA biotin-XX: the remainder of RNA was mixed with 25 μl of 10× biotinylation buffer (100 mM HEPES (pH 7.4), 10 mM EDTA) and 12.5 μl MTSEA biotin-XX (1 mg/ml in N,N-dimethylformamide) and incubated for 30 min at room temperature in the dark with gentle rotation (Duffy *et al*, 2015). Following completion of the labelling reaction, free biotin was removed by chloroform:isoamyl alcohol (24:1) extraction. Biotinylated RNA was captured with 60 μl of Dynabeads MyOne Streptavidin C1 (Invitrogen, 65001) according to the manufacturer's instructions (including steps required for RNA application). To remove any unbound nucleic acids, streptavidin:biotinylated RNA complexes were washed twice with 1× B&W buffer (5 mM Tris–HCl (pH 7.4), 0.5 mM EDTA, 1 M NaCl). RNA was released by cleaving the disulphide bond previously formed between 4-SU and MTSEA biotin-XX with 100 mM DTT at room temperature. Eluted RNA was precipitated with glycogen, 0.3 M NaCl and isopropanol overnight at −20°C, followed by high-speed centrifugation and 70% ethanol wash. RNA was resuspended in RNase-free water, converted to cDNA using QuantiTect RT kit (QIAGEN) and analysed with qPCR as previously described.

## DRIP-seq

Sample preparation for DRIP-seq was essentially performed as described above, but with some minor changes. All the samples were spiked (Orlando *et al*, 2014) with the same batch of *Drosophila* S2 cells in 1:4.2 ratio to DT40 cells. Digested DNA was treated with 100 μg RNase A in the presence of 0.5 M NaCl for 2 h at 37°C. Elution from magnetic beads was performed for 1 h at 37°C in 300 μl elution buffer supplemented with 0.1 mg/ml RNase A. To ensure complete elution, 10 μg Proteinase K was added and incubated for a further 90 min at 37°C. DNA was purified by phenol: chloroform:isoamyl alcohol extraction, quantified with Qubit dsDNA HS Assay Kit (Invitrogen), diluted with ultra-pure water to 55 μl and sheared with Covaris M220 Focused-ultrasonicator and Holder XTU to average size of 300 bp in microTUBE-50 AFA. DNA libraries were built using NEBNext Ultra II DNA Library Prep Kit (New England Biolabs) as per the manufacturer's instructions.

## RNA DIP-seq

Between 60 and 100 million human cells were spiked in with DT40 cells (1/10 ratio), harvested and washed with cold PBS and nuclei isolated by lysing cells in HLB + N [10 mM Tris–HCl (pH 7.5), 10 mM NaCl, 2.5 mM MgCl$_2$ and 0.5% (vol/vol) NP-40] and passing it through a 10% sucrose cushion. Collected nuclei were lysed overnight in NLB [25 mM Tris–HCl (pH 7.4), 1% SDS, 5 mM EDTA, 0.125 mg/ml Proteinase K] with agitation at 37°C. Nucleic acids were purified with 1 M potassium acetate (pH 5.5), precipitated and treated with 1 U of RNase I (Ambion, AM2294) per 90 μg of DNA (15 min at 37°C) to degrade soluble RNA. DNA was purified with

phenol:chloroform, diluted with the IP dilution buffer and sheared with a Bioruptor Plus (Diagenode) to average size of 300 bp. DNA:RNA hybrids were immunoprecipitated with S9.6 mAb (1 μg antibody for each 2 μg of DNA) overnight. Immunocomplexes were captured with Protein G beads and washed as for ChIP and DRIP preparation. DNA:RNA hybrids were eluted by incubating the sample with Proteinase K for 2 h at 42°C. Nucleic acids were cleaned up with phenol:chloroform:isoamyl alcohol, precipitated with glycogen and resuspended in water, denatured for 5 min at 90°C and immediately placed on ice. DNA moiety of DNA:RNA hybrids was removed with 4 U DNase I for 30 min at 37°C. RNA was extracted with QIAzol, precipitated overnight and dissolved in RNase-free water. Strand-specific Illumina-compatible libraries were prepared with NEBNext Ultra II Directional RNA Library Prep Kit (NEB, E7760) with 100 ng input. Libraries were quality-checked as before and sequenced on a NextSeq 500 (Illumina).

### Quantification, display and statistical analysis of deep sequencing data

DRIP-seq libraries were sequenced on an Illumina HiSeq 4000, and RNA-DIP libraries were sequenced on an Illumina NextSeq. Reads were trimmed and quality-filtered using Trim Galore (version 0.4.4; https://www.bioinformatics.babraham.ac.uk/projects/trim_galore/), and then aligned to genomes with bowtie2 (version 2.26; Langmead & Salzberg, 2012) using default settings. DT40 reads were aligned to Ggal 5.0 and Dmel r6.18, and BOBSC reads were aligned to GRCh38 and Ggal 5.0. Alignments were filtered for uniquely matching reads and separated into sample and spike-in. Peaks were called on filtered alignments using MACS2 (version 2.1.1.20160309) with the default settings and −g 1.87e9 (or −g hs for human)—broad (Feng *et al*, 2012). Peak heights were normalised to the read number of the spike-ins and compared using the Mann–Whitney *U*-test. Overlaps between peaks were calculated using bedtools2 closest (version 2.27.1) with default settings (Quinlan & Hall, 2010). Peaks were considered to be overlapping if at least 1 bp overlapped. Sequences with H-DNA-forming potential were identified with the Triplex R package (Hon *et al*, 2013). G4 motifs were identified using the Quadparser algorithm (Huppert & Balasubramanian, 2005) with the regex $[G_{3\text{-}5}N_{1\text{-}7}G_{3\text{-}5}N_{1\text{-}7}G_{3\text{-}5}N_{1\text{-}7}G_{3\text{-}5}]$. The .bed files containing the positions of the identified H-DNA- and G4-forming sequences have been deposited. Enrichment testing for secondary structures was performed using the hypergeometric test.

To generate the normalised profiles presented in Figs 5A and 6A, the number of uniquely mapped reads per 100-bp windows along Galgal5 genome was determined and normalised by the total number of uniquely mapped reads for each experiment. Values for wild type and *primpol* were then normalised to the relative abundance of the spiked genome (*Drosophila melanogaster* from S2 cells in the case of the DT40 DRIP-seq data and *Gallus gallus* from DT40 cells in the case of the BOBSC RNA DIP-seq data). For RNase H-treated controls (in which both the experimental and spike genome signal will be reduced), the read height was further corrected to reflect the amount of nucleic acid retrieved after immunoprecipitation, which was at least eightfold less following RNase H treatment. In the case of RNA DIP-seq, it was necessary to pool all RNase H-treated samples to obtain sufficient material to build a sequencing library.

## Data availability

Deep sequencing data have been deposited in the GEO repository (https://www.ncbi.nlm.nih.gov/geo/) with accession number GSE112747.

**Expanded View** for this article is available online.

## Acknowledgements
We would like to thank Maria Daly and her team in the LMB Flow Cytometry Facility for cell sorting, Toby Darling and Jake Grimmett of LMB Scientific Computing for assistance, Prof. David Leach (University of Edinburgh) for the gift of the SbcCD-deficient *E. coli* strain DL733, the CRUK Cambridge Institute Genomics Core for Illumina sequencing, Ludovic Vallier and the Wellcome Trust Sanger Institute for the BOBSC iPS line and Cara Eldridge for culturing the iPS cells. The generation of the *primpol* BOBSC line was carried out under the auspices of the COMSIG consortium, supported by a Wellcome Trust strategic award (101126/B/13/Z). Finally, we thank all members of the Sale Lab, Joe Yeeles, Martin Taylor and Pierre Murat for helpful discussions. S.Š. was funded by the LMB Cambridge International Scholarship and A.C. by a UKRI Innovation Fellowship. Work in the J.E.S. Lab is supported by a core grant from the MRC to LMB (U105178808). The A.J.D. Laboratory is supported by grants from the BBSRC: BB/H019723/1 and BB/M008800/1. T.A.G. was supported by a University of Sussex PhD studentship. The N.J.P. Lab is supported by a Wellcome Trust Investigator Award (107928/Z/15/Z) and an ERC Advanced Grant (339170).

## Author contributions
SŠ and JES conceived the study and wrote the paper with input from all authors. SŠ performed all the experiments. SMT-W and NJP developed and with SŠ performed the RNA DIP-seq experiments. TAG and AJD identified and created the PrimPol mutant cDNAs. AC and GG analysed the deep sequencing data.

## Conflict of interest
The authors declare that they have no conflict of interest.

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
