## [Review Process File · The EMBO Journal]

R-loop formation during S phase is restricted by PrimPol-mediated repriming

Saša Šviković, Alastair Crisp, Sue Mei Tan-Wong, Thomas A. Guilliam, Aidan J. Doherty, Nicholas J. Proudfoot, Guillaume Guilbaud and Julian E. Sale.

Review timeline:

Submission date:	8 th May 2018
Editorial Decision:	13 th June 2018
Revision received:	8 th October 2018
Editorial Decision:	25 th October 2018
Revision received:	29 th October 2018
Accepted:	6 th november 2018

Editor: Hartmut Vordermaier

Transaction Report:

1st Editorial Decision

13th June 2018

Thank you again for submitting your manuscript EMBOJ-2018-99793, "S phase R-loop formation is restricted by PrimPol-mediated repriming." We have in the meantime received a complete set of reviews from three referees, which you will find enclosed below for your information. Furthermore, the referees have also extensively cross-discussed the comments among themselves, coming to the overall conclusion that it would be worthwhile to further consider a revised manuscript answering their various concerns. Therefore, please start preparing a revision along the lines suggested by the referees. Since the referees had some continuing disagreement on the relevance and relative importance of some of the issues raised, I would at the same time like to invite you to compile and send a tentative letter of response to the referee comments already at the start of the revision period, so we could determine (in light of the referee cross-consultations and possibly in further discussions with some of the reviewers) how the various points might be best clarified, and which experimental additions would appear to be essential for this revision.

REFeree REPORTS.

Referee #1:

It was a real pleasure to read and review the manuscript from Šviković et al. The topic is very interesting and highly significant to a broad audience with an interest in genome stability. The experimental data is of high quality and the appropriate, rigorous controls have been performed, and the conclusions are logical and well supported by the presented data.

The authors showed that:

- a short purine-rich repeat, (GAA)₁₀, which normally doesn't have a significant impact on transcription or replication, is actually an impediment for replication in vivo, but the re-priming

activity of the Primase-Polymerase PrimPol "rescues" the processive replication at these short repeats

- the (GAA)₁₀ sequence causes epigenetic instability through this replication-dependent mechanism
- the replication-impeding effect of these short repeats is dependent on DNA:RNA hybrid formation
- overexpression of RNase H1 (even if limited only to S-phase) completely bypasses the requirement for PrimPol to restore processive replication, while stabilisation of R-loops enhances the replication-impeding effect of the (GAA)₁₀ repeats
- R-loops are significantly elevated in S-phase in PrimPol- cells, the majority of the elevated signal is around secondary structure-forming sequences

The lack of criticism and questions doesn't reflect the lack of careful consideration of the manuscript, but rather, I think the study is robust and convincing in its current form and I don't see the need for additional experiments before publication.

I have only one minor comment: Figure 1B Legend says "red outline negative (BU-1 knockout) control" - which I found confusing at first, but I assume it is only a Bu-1a knockout.

Referee #2:

Long tracts of (GAA)_n repeats are difficult to replicate and cause genomic instability, presumably because they are prone to form secondary structures and accumulate R-loops. In this manuscript, Saša Šviković and colleagues report the important observation that short tracts GAA trinucleotide repeats (10 to 20) can also induce replication fork stalling in the absence of PrimPol, a recently identified primase-polymerase involved in the repriming of DNA synthesis at stalled forks. This conclusion is based on the analysis of the expression instability at the BU-1 locus in chicken DT40 cells, a stochastic and replication-dependent event that has been extensively characterized by the Sale lab over the past few years. Remarkably, this instability is orientation dependent and is largely suppressed upon overexpression of RNase H1, suggesting that (GAA)₁₀₋₂₀ repeats block leading strand synthesis in a manner that depends on the formation of RNA:DNA hybrids. To address this attractive possibility, the authors have performed an extensive analysis of RNA:DNA hybrids at the BU-1 locus and at the genome wide level, both in DT40 and human cells. Based on these observations, they propose an original model involving the formation of a triplex between the RNA:DNA hybrid on the lagging strand and GAA sequences on the leading strand. This model is attractive and it is fully consistent with the fluctuation analysis for generation of Bu-1 loss variants. It has major implications for our understanding of the instability of trinucleotide repeats and should therefore be of wide general interest. However, I have major concerns regarding the design and the interpretation of experiments on RNA:DNA hybrids. In my opinion, further work is needed to support the view that RNA:DNA hybrids are involved in the replication impediments mediated by short GAA tracts.

Major issues:

1. Using DRIP-qPCR, the authors show that RNA:DNA hybrids are present at the BU-1 locus (Fig. 1A), and especially at the TTS, which is a common feature of many Pol2 genes. They also show that the signal around (GAA)_n increases in the absence of PrimPol. However, this signal extends quite far on both sides of the (GAA)_n and is at least 20 times lower than the signal detected at the TTS, which is surprisingly insensitive to the *in vivo* expression of RNaseH1 (Fig. 1C). These results and their interpretation raise a number of questions. If leading strand synthesis is blocked by (GAA)_n, why would R-loop form both upstream and downstream of GAA? Moreover, what is the evidence that these hybrids are formed on the coding strand, opposite to the Purine-rich strand, as depicted on the model of Fig. 7. Finally, the amount of RNA:DNA hybrids around (GAA)_n is very low compared to TTS. This raises the question of why they are not removed by endogenous RNaseH1. One possible explanation is that these hybrids are present in only a subset of cells, when the locus is replicated, whereas R-loops form at the TTS throughout the cell cycle. This view is supported by the experiment shown Fig. 5. However, the TTS was not analyzed in this experiment and the increase of RNA:DNA hybrid levels in mid-S is rather modest compared to G1-arrested cells. It would be important to analyze the TTS of BU-1, but also other loci that do not contain GAA repeats, to clarify this issue.

2. The result shown in Fig. 5B is very surprising. A large body of evidence indicate that R-loops

form in a cotranscriptional manner and cover a large fraction of the genome. Since most of the genome is transcribed throughout the cell cycle, it is very unlikely that the sharp transitions observed in both WT and primpol cells are real. I understand that the authors used an alternative approach to DRIP to quantify R-loops, but to provide convincing evidence that this signal correspond to R-loop, they need to show that it is resistant to RNase III and sensitive to RNase H. They also need to perform the experiment several times and show the quantification of at least three biological replicates, after normalization to ssDNA. Ideally, they could also perform a classical DRIP experiment and quantify the total signal on a slot blot, in order to compare the two techniques.

3. The results of the DRIP-seq experiments presented in Fig. 6 are again difficult to assess as only highly processed data are shown. It would be important to show an example of the distribution of the signal at representative regions and on a metagene, to see if the data recapitulate published profiles with R-loop enrichment at TSS, gene body and TTS. It is indeed surprising that only 41% of the R-loop peaks overlap with genes. If so, what do the other peaks correspond to? And if PrimPol only affects R-loop formation at very specific loci during phase, how to explain this global increase of R-loop signals in asynchronous cells? From this respect, the assumption that PrimPol does not have a direct role in processing R-loops as primpol cells are not sensitive to CPT (p. 7) is a very indirect argument and should be toned down. Overall, whether PrimPol has a role outside of S phase to remove R-loops is not clearly addressed here and this critical point needs to be clarified.

Minor issues:

1. The model presented in Fig. 1A suggests that uncoupling between helicase and leading strand polymerase extends up to 4 kb ahead of the (GAA)_n motif, which would be required to disrupt epigenetic information at the promoter region. I assume that this refers to a mutant situation, but this is not indicated. The authors should also discuss published evidence indicating that such an extensive uncoupling occurs in vivo. Finally, they should discuss the nature of the activity that could be responsible for repriming in the absence of PrimPol.

2. Figure 2: According to the authors, the "expression of catalytically inactive PrimPol (hPrimPol[AxA]) and the DNA-binding zinc finger mutant (hPrimPol[ZfKO]) shows that neither is able to prevent instability of BU-1 expression". Yet, this instability was largely reduced compared to the absence of primpol or the delta RBMA mutant. Does this reflect the persistence of a residual PrimPol activity in these mutants? This issue needs to be discussed.

3. Fig. 3F: It is surprising that the effect of Gg RNase H1 #6 is not statistically significant. Is one-way ANOVA the right test to apply here? Have the authors tested the normality of the distributions?

Referee #3:

PrimPol is known to reprime replication to bypass DNA damage, including UV damage, AP sites, G-quadruplexes, in both nuclear and mitochondrial DNA. This manuscript by Sale and collaborators extends the spectrum of structures at which PrimPol can initiate repriming to include R-loops. This is an interesting result, but it is studied largely in the context of a model reporter in chicken DT40 cells. The significance would be enhanced by expansion of the genomewide analysis in human cells, with more information about the specific short repetitive tracts or other motifs that may depend upon PrimPol.

Most of the manuscript (Figures 1-5) is devoted to a meticulous and detailed analysis of the how replication processivity is affected by insertion of (GAA)₁₀ or (GAA)₂₀ repeats at the BU-1 locus in chicken DT40 B cells. Only near the end is the important larger question addressed: does PrimPol prime re-initiation at R-loops in other genes and in human cells. The genomewide analyses of these two organisms are the most significant component of the manuscript, but they are presented together, in a single figure (Figure 6), without specific examples, detail or depth.

PrimPol function is documented in assays of replication processivity at the BU-1 locus in chicken

DT40 B cells, the same approach as used previously to demonstrate that PrimPol restarts leading strand replication stalled by quadruplex structures (Schiavone et al. 2016). At the BU-1 locus, gene expression is downregulated when the BU-1 gene loses activating marks due to replication stalling at a naturally-occurring G4 motif 3.5 kb downstream of the TSS. Tallying cells that have lost surface BU-1 expression by flow cytometry thus provides a very convenient surrogate assay for the frequency with which cells experience interrupted DNA synthesis at BU-1. Substitution of long GAA tracts (30-75 repeats) for the natural G4 motif causes global downregulation of expression, presumably by accumulation of nascent RNA in chromatin, while substitution of (GAA)₁₀ or (GAA)₂₀ repeats causes stochastic loss of BU-1 expression. Accumulation of loss variants requires that the GAA repeat be on the leading strand, and is greatly exacerbated by ablation of PrimPol. Suppression of epigenetic instability by PrimPol depends on its catalytic activity and DNA binding zinc finger motif.

What causes arrest? The authors hypothesize that it is an R-loop formed by the transcribed GAA repeat. To test this, RNA/DNA hybrids were assayed across the locus by ChIP with the S9.6 antibody, which recognizes RNA/DNA hybrids. This reveals that the presence of a (GAA)₁₀ repeat results in an enhanced DRIP signal in primpol-deficient (GAA)₁₀ cells. However, enhancement is evident throughout the gene, and is not localized to the 3.5 region as one might have thought (Fig. 3A), even though a very strong signal is localized to the very 3' end where RNA/DNA hybrids promote transcription termination. This is surprising. Could the signal derive in part from nascent chromatin-associated RNAs that overwhelmed BU-1 expression at high repeat numbers (Fig. 1B)? This could be tested by comparing signals from BU-1 loci with different numbers of repeats (from zero up). These results raise the concern that the S9.6 antibody may not be completely specific for RNA/DNA hybrids, but may cross-react with chromatin-bound nascent RNAs.

The literature includes relatively few controls for specificity or cross-reactivity of the widely-used S9.6 antibody. Controls should be included showing that S9.6 specifically IPs RNA/DNA hybrids and not chromatin-associated RNA in DT40 cells; and accompanied by controls for specificity of this antibody in human cells used for genome-wide analysis.

Taking a different tack to identify the source of epigenetic instability, treatment with RNase H1 was shown to abrogate accumulation of BU-1 loss variants (Fig. 3F). The converse experiment, expression of the DNA:RNA hybrid binding domain (HBD) of RNaseH1, caused increased accumulation of loss variants (Fig 4). These results of RNaseH1 and RNaseH1 HBD expression are consistent with the view that RNA/DNA hybrids are in some way responsible for accumulation of loss variants. However, they do not show that the structures form at the BU-1 locus itself, and leave open the possibility RNaseH1 treatment has improved expression or activity of some factor that rescues BU-1 expression.

The hybrids formed in the absence of PrimPol are correlated with replication by experiments showing that their abundance increases in S phase, as measured both using the antibody and by metabolic labeling of nascent RNA. As presented, the results of these different analyses (Fig. 5B and Fig. EV6) seem rather different, with the mid-S phase peak in Fig. 5B not evident in Fig. EV6; and the only clear distinction between wild type and primpol-deficient cells in G2 phase in the latter figure. Is there a real difference? These experiments and results need better explanation.

The association between R-loops and PrimPol-dependent replication restart genome-wide is determined by DRIP-Seq comparing recovery of RNA-DNA hybrids from primpol-deficient and wild type DT40 cells using the S9.6 antibody. The great majority of peaks were common between these two backgrounds, but peak heights were significantly greater in the primpol-deficient cells, suggesting that primpol-deficiency results in a higher steady state level of R-loops, but not induction of new R-loops. This could mean that PrimPol preferentially functions at a subset of genes. This could reflect enrichment of the enzyme's preferred priming site (3'GTCC5': Garcia-Gomez et al MolCell 2013), and even though this site is short it would be useful to score its enrichment/depletion near peaks.

To ask if the peaks identified correlate with possible R-loop formation, enrichment of H-DNA motifs was scored within peaks. H-DNA motifs can correlate with regions with potential to form R-loops, but not enough detail or precedent is presented to rationalize and ground this search. To keep this from seeming like a sleight of hand, more information is required to enable the reader to

evaluate the details of the search and especially its stringency. The algorithm needs be stated explicitly and put into context. How long are H-DNA motifs, and how many H-DNA motifs are there in the chicken and human genomes? Has the algorithm used to identify H-DNA motifs been validated in other genomewide analyses? What happens if more (or less) stringent criteria are applied to H-motif identification?

Other information should be provided in order to evaluate the genomewide results: (1) peak width, which determines the potential for overlaps; (2) maps of peak distribution across representative individual genes, identifying sites with potential for formation of R-loops, G4 DNA, etc along the genes - this would substantiate claims of the sort found in the Discussion that PrimPol reprimates at short tandem repeats throughout the genome.

Very high statistical significance is reported in both DT40 chicken B cells and the human BOBSC stem cell line in Figure 6, but the labels in the figure and the details of the analysis are somewhat unclear. Is the presence of a single H-DNA motif in a long gene sufficient for it to score in the statistical analysis?

Major comments:

1. Include controls for S9.6 antibody specificity in DT40 and human BOBSC stem cells.
2. DRIP-Seq peak widths must be presented to understand the significance of overlaps. It would be useful to see how results change as peak widths are narrowed; and with changes in stringency of the H-motif algorithm.

The Introduction and Discussion over-emphasize short repetitive tracts, considering that only one specific example is shown, the GAA repeat cloned into the BU-1 locus.

Minor comments:

Discussion: "We now show that the repriming function of PrimPol is frequently deployed at short tandem repeats throughout the genome...." This has not been shown and must be rephrased. Demonstrating this could be done with more detailed presentation of the genomewide analysis, which would be appropriate and interesting.

Fig. 3E: Overlay cell cycle profile on data

Figures 2 and 4 might be presented as supplementary data rather than in the main text. This would give space for deeper analysis of the genomewide results.

Fig. EV6: Numbers in flow cytometry output quadrants should be large enough to read.

1st Revision - authors' response

8th October 2018

EMBOJ-2018-99793

S phase R-loop formation is restricted by PrimPol-mediated repriming

Our response (blue) to the referees' comments (black)

Referee #1:

It was a real pleasure to read and review the manuscript from Šviković et al. The topic is very interesting and highly significant to a broad audience with an interest in genome stability. The experimental data is of high quality and the appropriate, rigorous controls have been performed, and

the conclusions are logical and well supported by the presented data.

The authors showed that:

- a short purine-rich repeat, (GAA)₁₀, which normally doesn't have a significant impact on transcription or replication, is actually an impediment for replication in vivo, but the re-priming activity of the Primase-Polymerase PrimPol "rescues" the processive replication at these short repeats
- the (GAA)₁₀ sequence causes epigenetic instability through this replication-dependent mechanism
- the replication-impeding effect of these short repeats is dependent on DNA:RNA hybrid formation
- overexpression of RNase H1 (even if limited only to S-phase) completely bypasses the requirement for PrimPol to restore processive replication, while stabilisation of R-loops enhances the replication-impeding effect of the (GAA)₁₀ repeats
- R-loops are significantly elevated in S-phase in PrimPol- cells, the majority of the elevated signal is around secondary structure-forming sequences

The lack of criticism and questions doesn't reflect the lack of careful consideration of the manuscript, but rather, I think the study is robust and convincing in its current form and I don't see the need for additional experiments before publication.

We are naturally delighted by these positive comments and thank the referee for the clear summary of the key findings of our study.

I have only one minor comment: Figure 1B Legend says "red outline negative (BU-1 knockout) control" - which I found confusing at first, but I assume it is only a Bu-1a knockout.

This is correct. The red line represents the FACS profile of cells in which Bu-1a has been genetically inactivated. We have clarified this in the legend.

Referee #2:

Long tracts of (GAA)_n repeats are difficult to replicate and cause genomic instability, presumably because they are prone to form secondary structures and accumulate R-loops. In this manuscript, Saša Šviković and colleagues report the important observation that short tracts GAA trinucleotide repeats (10 to 20) can also induce replication fork stalling in the absence of PrimPol, a recently identified primase-polymerase involved in the repriming of DNA synthesis at stalled forks. This conclusion is based on the analysis of the expression instability at the BU-1 locus in chicken DT40 cells, a stochastic and replication-dependent event that has been extensively characterized by the Sale lab over the past few years. Remarkably, this instability is orientation dependent and is largely suppressed upon overexpression of RNase H1, suggesting that (GAA)₁₀₋₂₀ repeats block leading strand synthesis in a manner that depends on the formation of RNA:DNA hybrids. To address this attractive possibility, the authors have performed an extensive analysis of RNA:DNA hybrids at the BU-1 locus and at the genome wide level, both in DT40 and human cells. Based on these observations, they propose an original model involving the formation of a triplex between the RNA:DNA hybrid on the lagging strand and GAA sequences on the leading strand. This model is attractive and it is fully consistent with the fluctuation analysis for generation of Bu-1 loss variants. It has major implications for our understanding of the instability of trinucleotide repeats and should therefore be of wide general interest. However, I have major concerns regarding the design and the interpretation of experiments on RNA:DNA hybrids. In my opinion, further work is needed to

support the view that RNA:DNA hybrids are involved in the replication impediments mediated by short GAA tracts.

Major issues:

1. Using DRIP-qPCR, the authors show that RNA:DNA hybrids are present at the BU-1 locus (Fig.1A), and especially at the TTS, which is a common feature of many Pol2 genes. They also show that the signal around (GAA)_n increases in the absence of PrimPol. However, this signal extends quite far on both sides of the (GAA)_n and is at least 20 times lower than the signal detected at the TTS, which is surprisingly insensitive to the in vivo expression of RNaseH1 (Fig. 1C). These results and their interpretation raise a number of questions. If leading strand synthesis is blocked by (GAA)_n, why would R-loop form both upstream and downstream of GAA?

Previous genome wide analyses of R-loop formation have indicated that about a quarter of human genes exhibit so called “sticky” behaviour i.e. they are prone to accumulate R-loops across their body (Sanz et al., 2016). This tendency is observed mostly in very short (<10 kb) and highly expressed genes. The R-loop formation across *BU-1* that we observe with DRIP-qPCR, DRIP-seq and RNA-DIP (see below) is consistent with this behaviour. We suggest that the formation of a local DNA:RNA hybrid induced by the (GAA)_n repeat sequence may act to nucleate DNA:RNA hybridisation throughout the gene body, as previously suggested (Roy et al., 2008). We have added a sentence noting this observation and the potential explanation.

Moreover, what is the evidence that these hybrids are formed on the coding strand, opposite to the Purine-rich strand, as depicted on the model of Fig. 7.

During our human RNA DIP-seq experiments we included a DT40 spike-in control. We have now analysed and expanded this data to examine the strandedness of R-loop formation in *BU-1*. We observe a strong signal corresponding to the RNA component of DNA:RNA hybrids in the *BU-1* locus overwhelmingly in the direction of gene transcription (new Appendix Fig S3). This firmly places the DNA:RNA hybrid on the lagging strand and opposite the purine-rich strand consistent with the early biochemical studies of Grabczyk and colleagues (Grabczyk & Fishman, 1995, Grabczyk et al., 2007, Grabczyk & Usdin, 2000), which demonstrated that an R-loop would only form when purine-rich DNA is transcribed as a coding (non-template strand). This analysis also confirms that the signal is spread across the whole gene. We also confirmed that the signal represents bona fide DNA:RNA hybrids as it is lost by pre-treatment of the isolated nucleic acids with recombinant *E.coli* RNase H prior to S9.6 immunoprecipitation and high-throughput sequencing (new Appendix Fig S3). We have noted this observation in the Results section of the revised manuscript.

Finally, the amount of RNA:DNA hybrids around (GAA)_n is very low compared to TTS. This raises the question of why they are not removed by endogenous RNaseH1. One possible explanation is that these hybrids are present in only a subset of cells, when the locus is replicated, whereas R-loops form at the TTS throughout the cell cycle. This view is supported by the experiment shown Fig. 5. However, the TTS was not analyzed in this experiment and the increase of RNA:DNA hybrid levels in mid-S is rather modest compared to G1-arrested cells. It would be important to analyze the TTS of *BU-1*, but also other loci that do not contain GAA repeats, to clarify this issue.

We have now repeated the cell cycle DRIP analysis from scratch to include the TTS and all data. As we originally observed, the signal at the TTS is higher than that in the gene body. We can now see, as predicted by the referee, that in *primpol* cells the DRIP signal increases markedly shortly after the cells enter S phase, coincident with the time that we have previously reported the locus to be

replicated. This similar behaviour to the gene body around the +3.5 (GAA) repeat prompted us to look more closely at the sequence around the TTS. This revealed a cluster of structure-forming sequences, including two prominent G4 motifs and a series of triplex-forming motifs (new Appendix Fig S4). These are too far away from the promoter to cause epigenetic instability of the gene according to our model. Indeed, we have shown this by removing the structure-forming sequence at the +3.5 position (Figure 1C and Schiavone et al., 2014, Schiavone et al., 2016). However, these sequences should be capable of structure formation and may account for the increase in R-loop formation around the +11.5 amplicon in mid S-phase in the absence of PrimPol. Given the large proportion of cells in S phase in an asynchronous culture of DT40 (Fig EV4), their presence is likely to account for the increase in the steady state +11.5 DRIP signal we report in Figure 3. We have now also shown that an increase in the TTS R-loop signal is not a general feature, as it is not observed in a selection of loci that do not harbour any identifiable structure-forming DNA in the vicinity of their TTS (new Appendix Fig S7). It will be interesting to explore in future work whether DNA structure formation plays any role in transcription termination, or whether the increased R-loop signal at these sites in *primpol* cells simply reflects the additive effect of transcription termination and replication stalling, an explanation we currently favour.

Concerning the differential effects of over expression of RNase H1. The factors that determine which R-loops are susceptible and which are resistant to removal by *in vivo* over expression of RNaseH1 remain unclear. R-loop levels at any given site reflect a balance between formation and disassembly, with a modest shift toward disassembly expected when RNaseH1 is ectopically overexpressed. We favour the explanation offered by the referee that since the R-loops associated with transcription termination are constitutively formed during the cell cycle, they are less susceptible to complete removal by increased endogenous RNase H1 expression. As they may also be necessary for cell fitness, the anticipated higher levels of RNase H1 needed to remove them may be selected against. In contrast, as we show clearly in the new Fig 4, the R-loops formed during replication and associated with secondary structures are evanescent, and likely serve no useful function. We suggest that this makes them more susceptible removal by increased RNase H1 overexpression. These hypotheses remain to be explored in more depth.

2. The result shown in Fig. 5B is very surprising. A large body of evidence indicate that R-loops form in a cotranscriptional manner and cover a large fraction of the genome. Since most of the genome is transcribed throughout the cell cycle, it is very unlikely that the sharp transitions observed in both WT and *primpol* cells are real. I understand that the authors used an alternative approach to DRIP to quantify R-loops, but to provide convincing evidence that this signal correspond to R-loop, they need to show that it is resistant to RNase III and sensitive to RNase H. They also need to perform the experiment several times and show the quantification of at least three biological replicates, after normalization to ssDNA. Ideally, they could also perform a classical DRIP experiment and quantify the total signal on a slot blot, in order to compare the two techniques.

It turns out that the suspicions of the referee about this experiment have an unanticipated foundation. One point of this experiment was indeed attempted to develop an approach that would allow us to monitor R-loop formation independently of S9.6. At the suggestion of the referee we set up experiments aimed at systematically addressing the effects of the different RNase treatments on the readout. The experiments included not only treatment with RNase HI and RNase III, but also dropping out the treatment of the restriction digested nucleic acids with RNase A. We had previously added this to all conditions to remove contaminating signal from double stranded RNA, which can be detected by S9.6 and interfere with DNA:RNA hybrid detection. Unexpectedly, omitting RNase A resulted in abrogation of the slot blot signal (Figure 1, below). Further, when RNase A is present, the slot blot signal is not sensitive to RNase H1. Further investigation leads us to believe that the signal we are detecting in the slot blot actually reflects a fraction of DNA bound

RNase A that survives the phenol/chloroform step and subsequently becomes biotinylated by the thiol-specific EZ-link HPDP-Biotin reagent. RNase A is known to bind DNA despite not cutting it, and indeed binds more strongly to single stranded regions (Felsenfeld et al., 1963). More recently, the problem of RNase A binding to DNA and surviving phenol:chloroform extraction has been noted as an explanation for DNA depletion following RNase A treatment (Dona & Houseley, 2014). It is curious that the signal we detect in many ways parallels the results we see with S9.6, for instance the consistent increase in signal in PrimPol-deficient cells suggests that the assay could be picking up increased single stranded DNA in these cells. However, as we cannot currently be confident that this is the case, we have therefore decided to remove the original Figure 5B from the paper.

Figure 1. RNase A induces artefactual detection of 4-SU labelled and biotinylated RNA by slot blot. See text for discussion.

However, we have now modified the assay so that we detect the 4-SU labelled RNA by qRT-PCR and we have applied this to the *BU-1* locus (new Fig 4D). We have shown that the signal we detect is entirely sensitive to pre-treatment with RNase H1, giving us confidence that we are detecting labelled RNA that is forming part of a DNA:RNA hybrid (new Fig 4E). The results of applying this approach to the different stages of the cell cycle, with three completely independent synchronisations and replicates of the experiment, are striking. The amount of material recovered from each experiment limits the number of sites that can be interrogated and so we have confined our analysis to the region around the $(GAA)_{10}$ repeat. The assay reveals a spike in R-loop production around the $(GAA)_{10}$ repeat in *primpol*, but not wild type, cells at exactly the time we have previously determined the locus to be replicated, providing S9.6-independent evidence of R-loop synthesis at a structure-forming sequence when repriming is defective (new Figs 4F & G).

3. The results of the DRIP-seq experiments presented in Fig. 6 are again difficult to assess as only highly processed data are shown. It would be important to show an example of the distribution of the signal at representative regions and on a metagene, to see if the data recapitulate published profiles with R-loop enrichment at TSS, gene body and TTS. It is indeed surprising that only 41% of the R-loop peaks overlap with genes.

We now expanded the genome wide analysis in the chicken human cells to occupy the whole of the new Figures 5 and 6. We present example normalised plots of the S9.6-detected R-loop signal across representative genes in wild type and PrimPol-deficient cells (new Figs 5A & 6A). The wild type and *primpol* signals are normalised to the spike-in controls and visually make two important points, which were not as evident in the original submission. First that the pattern of enrichment is highly correlated between wild type and *primpol* and second, the peaks in *primpol* are generally higher than wild type, as confirmed in the genome wide analysis presented below. We also now explicitly show the effects of pre-treatment of the samples with RNase H, confirming the specificity of S9.6 in this context.

We now present a metagene analysis of R-loop distribution (new Figs 5B & 6B) to aid comparison between our data and previously published datasets. We note that DRIP-Seq has not previously been performed in avian cells. Nonetheless, our analysis reveals a comparable pattern of R-loops enrichment at gene promoters and termini in our chicken dataset and the previously published human DRIP-seq datasets in which approximately 35% gene bodies overlap with R-loops (Ginno et al., 2012, Sanz et al., 2016). Further, the number of detected R-loop peaks is in the same range (~35000) as in our study.

It should be noted that DRIP-Seq approaches (i.e. sequencing of the DNA moiety of DNA:RNA Hybrids) have generally poorer correlation with the genic regions as opposed to when the RNA moiety is sequenced (DRIPc-Seq for example). If so, what do the other peaks correspond to?

This is indeed also the case in our datasets: there is much less intergenic signal in the human RNA DIP-seq data than in the DT40 DRIP-seq, which is generally much noisier. We have now examined the association of the peaks in the DRIP-seq data with underlying structures and have found that 19% of extragenic peaks are within 1kb of an H-DNA sequence in wild type cells, 17% in *primpol*. We have included a sentence noting this in the revised manuscript. It is important to note that not all peaks are associated with the limited range of secondary structure-forming sequences we have scored (H-DNA and a subset of potential G4 motifs) or that all potential structures are associated with DRIP peaks.

And if PrimPol only affects R-loop formation at very specific loci during phase, how to explain this global increase of R-loop signals in asynchronous cells?

The global increase in steady state R-loops in *primpol* cells is consistent, but relatively modest, in the order of 1.2 – 1.3-fold. We believe that this can readily be explained by the increase in R-loop formation during S-phase that we report. Since some 50 – 60% of cells in an asynchronous culture of both wild type and *primpol* DT40 are in S phase (see Figure EV4), an increase of R-loop levels in cells during S-phase is sufficient to explain the increase in gross signal in an asynchronous culture. As the genome-wide analysis shows, the global increase in steady state R-loop levels in *primpol* cells can be accounted for by H-DNA and G4 containing genes.

From this respect, the assumption that PrimPol does not have a direct role in processing R-loops as *primpol* cells are not sensitive to CPT (p. 7) is a very indirect argument and should be toned down. Overall, whether PrimPol has a role outside of S phase to remove R-loops is not clearly addressed here and this critical point needs to be clarified.

We have toned this point down and have discussed this issue more fully. We agree that this is an important point. We cannot absolutely exclude that PrimPol additionally acts outside S phase to control R-loops. However, we are not aware of any studies as yet providing clear support for PrimPol playing a significant role outside of S phase. Rather, all the existing evidence points to PrimPol operating during DNA replication as a lesion bypass polymerase or, likely much more

importantly, as a primase repriming after a range of replication impediments including DNA damage (Mouron et al., 2013), chain terminating nucleotides (Kobayashi et al., 2016) and secondary structures (Schiavone et al., 2016). Further, the data we present, particularly the measurements of R-loop synthesis, point to PrimPol preventing the formation of R-loops during S phase. However, even if PrimPol does play an as yet undetected function outside of DNA replication, this would not alter any of the conclusions we draw in this current paper concerning the importance of PrimPol in dealing with R-loops encountered during S phase. We have discussed this point more fully.

Minor issues:

1. The model presented in Fig. 1A suggests that uncoupling between helicase and leading strand polymerase extends up to 4 kb ahead of the (GAA)_n motif, which would be required to disrupt epigenetic information at the promoter region. I assume that this refers to a mutant situation, but this is not indicated.

Yes, this is correct and we have ensured that this is clear. The frequency of such events in wild type cells appears to be extremely low, below the threshold of detection by the Bu-1 fluctuation assay. We have additionally observed increased instability of *BU-1* expression, indicative of an increased frequency of helicase-polymerase uncoupling, under a number of circumstances, for example loss of DNA structure-selective helicases and polymerases (e.g. FANCI and REV1) (Sarkies et al., 2012, Schiavone et al., 2014), depletion of nucleotide pools (Papadopoulou et al., 2015) and stabilisation of G-quadruplexes by G4-binding ligands (Guilbaud et al., 2017).

The authors should also discuss published evidence indicating that such an extensive uncoupling occurs in vivo.

We have added discussion of this point, with references.

Finally, they should discuss the nature of the activity that could be responsible for repriming in the absence of PrimPol.

Currently, only two primases are known in vertebrates, PrimPol and the PRIM1 subunit of DNA polymerase α . While the latter may possibly contribute, recent biochemical experiments with the reconstituted yeast replisome suggests that leading strand repriming by PRIM1 is inefficient (Taylor & Yeeles, 2018). However, a number of other activities may exhibit functional redundancy with PrimPol-dependent repriming, including fork convergence. We have added some discussion of this point.

2. Figure 2: According to the authors, the "expression of catalytically inactive PrimPol (hPrimPol[AxA]) and the DNA-binding zinc finger mutant (hPrimPol[ZfKO]) shows that neither is able to prevent instability of BU-1 expression". Yet, this instability was largely reduced compared to the absence of primpol or the delta RBMA mutant. Does this reflect the persistence of a residual PrimPol activity in these mutants? This issue needs to be discussed.

We have previously reported that expression of the PrimPol ZfKO or AxA mutants compromises cell growth, likely through a dominant negative effect (Schiavone et al., 2016). Expression of these transgenes is unstable and a measurable proportion of cells lose them over the course of two to three weeks in culture, as assayed by flow cytometry of YFP-tagged proteins in individual clones. Those cells that retain the constructs exhibit some growth disadvantage compared with cells that retain expression resulting in their completing fewer cell cycles during the course of the experiment. This reduces the potential for the generation of Bu-1 loss variants. We compensate for this as far as

possible by assessing *BU-1* expression only in cells that have retained YFP. However, it is likely that these cells have been through fewer divisions than the full *primpol* knockout giving them less opportunity to accumulate Bu-1 loss variants. The observation that these mutant constructs do not fully complement the *BU-1* expression instability of *primpol* cells is consistent with the repriming function of PrimPol being required. The data we present for the ZfKO and AxA mutants at (GAA)₁₀ are comparable with our previously published observations at a G4 (Schiavone et al., 2016). Importantly, the RBM-A mutant, which does not exhibit any problems with transgene stability has now provided significant additional support that ssDNA-induced repriming plays a key role at (GAA)₁₀. We have clarified this point in the revised manuscript.

3. Fig. 3F: It is surprising that the effect of Gg RNase H1 #6 is not statistically significant. Is one-way ANOVA the right test to apply here? Have the authors tested the normality of the distributions?

Originally, we used ANOVA to perform multiple comparisons across the whole dataset. Most of the distributions from the fluctuation analyses meet tests of normality, but this is not invariably the case. We have therefore also performed the non-parametric Kruskal-Wallis test, which is now shown. An additional point is the question of the correct comparator in the experiment. The expression of the YFP-RNase H1-geminin degen construct is not as stable as the regular YFP-RNase H1 we use elsewhere in the study and in some clones can be lost during the course of the fluctuation analysis (Figure 2, below). It is therefore possible to separate cells that have lost the ability to express YFP during S phase from those that can. Those clones that have lost the construct therefore provide a good internal control for the experiment as they are unable to suppress the (GAA)₁₀-induced instability. We have now applied this analysis to the experiment and show this, along with the K-W test statistics, in Fig 3B of the revised manuscript.

Figure 2. Loss of the YFP-RNase H1-geminin degen construct during the course of a fluctuation analysis experiment. The red arrow indicates cells in an example well that have lost the construct by the end of the fluctuation analysis.

Referee #3:

PrimPol is known to reprime replication to bypass DNA damage, including UV damage, AP sites, G-quadruplexes, in both nuclear and mitochondrial DNA. This manuscript by Sale and collaborators extends the spectrum of structures at which PrimPol can initiate repriming to include R-loops. This is an interesting result, but it is studied largely in the context of a model reporter in chicken DT40 cells. The significance would be enhanced by expansion of the genomewide analysis in human cells, with more information about the specific short repetitive tracts or other motifs that may depend upon PrimPol.

Most of the manuscript (Figures 1-5) is devoted to a meticulous and detailed analysis of the how replication processivity is affected by insertion of (GAA)₁₀ or (GAA)₂₀ repeats at the BU-1 locus in chicken DT40 B cells. Only near the end is the important larger question addressed: does PrimPol prime re-initiation at R-loops in other genes and in human cells. The genome-wide analyses of these two organisms are the most significant component of the manuscript, but they are presented together, in a single figure (Figure 6), without specific examples, detail or depth.

We thank the reviewer for their appreciation of the genome wide analysis, and agree that it adds weight to our detailed genetic analysis. As discussed in the response to Referee 2, we have expanded our analysis of the genome wide data to include example traces for genomic regions and specific genes (normalised to the spike-in control) as well as a metagene analysis demonstrating the distribution of R-loop peaks. A very detailed analysis of the nature of the short repetitive tracts or other motifs that may depend on PrimPol is beyond the technical scope of this paper as the resolution of the techniques is not sufficiently high and additional information, such as the distribution of leading and lagging strand synthesis would be needed.

PrimPol function is documented in assays of replication processivity at the BU-1 locus in chicken DT40 B cells, the same approach as used previously to demonstrate that PrimPol restarts leading strand replication stalled by quadruplex structures (Schivone et al. 2016). At the BU-1 locus, gene expression is downregulated when the BU-1 gene loses activating marks due to replication stalling at a naturally-occurring G4 motif 3.5 kb downstream of the TSS. Tallying cells that have lost surface BU-1 expression by flow cytometry thus provides a very convenient surrogate assay for the frequency with which cells experience interrupted DNA synthesis at BU-1. Substitution of long GAA tracts (30-75 repeats) for the natural G4 motif causes global downregulation of expression, presumably by accumulation of nascent RNA in chromatin, while substitution of (GAA)₁₀ or (GAA)₂₀ repeats causes stochastic loss of BU-1 expression. Accumulation of loss variants requires that the GAA repeat be on the leading strand, and is greatly exacerbated by ablation of PrimPol. Suppression of epigenetic instability by PrimPol depends on its catalytic activity and DNA binding zinc finger motif.

What causes arrest? The authors hypothesize that it is an R-loop formed by the transcribed GAA repeat. To test this, RNA/DNA hybrids were assayed across the locus by ChIP with the S9.6 antibody, which recognizes RNA/DNA hybrids. This reveals that the presence of a (GAA)₁₀ repeat results in an enhanced DRIP signal in primpol-deficient (GAA)₁₀ cells. However, enhancement is evident throughout the gene, and is not localized to the 3.5 region as one might have thought (Fig. 3A), even though a very strong signal is localized to the very 3' end where RNA/DNA hybrids promote transcription termination. This is surprising.

See reply to Reviewer #2 (point 1), above, in which we discuss this issue in detail.

Could the signal derive in part from nascent chromatin-associated RNAs that overwhelmed BU-1 expression at high repeat numbers (Fig. 1B)? This could be tested by comparing signals from BU-1 loci with different numbers of repeats (from zero up). These results raise the concern that the S9.6 antibody may not be completely specific for RNA/DNA hybrids, but may cross-react with chromatin-bound nascent RNAs.

We did report increasing chromatin-associated RNA as a function of increased (GAA)_n tract length (Fig EV3), likely reflecting the formation of R-loops and stalling of the RNAPII as previously suggested (Groh & Gromak, 2014, Punga & Buhler, 2010), although at $n = 10$ this increase is very slight. However, in all our DRIP experiments we have treated the isolated nucleic acids with RNase A to degrade any contaminating single stranded RNA species, including those originating from chromatin associated RNA. All the signals reported in our DRIP-qPCR experiments were normalised to samples treated with RNase H, but we now formally show that the signal is sensitive to RNase H and insensitive to RNase III (new Appendix Fig S2). The enrichments we observe should thus reflect only the *bona fide* R-loop signal and we are therefore as confident as we can be that the signal is specific is not accounted for by any accumulation of chromatin associated RNA.

The literature includes relatively few controls for specificity or cross-reactivity of the widely-used S9.6 antibody. Controls should be included showing that S9.6 specifically IPs RNA/DNA hybrids and not chromatin-associated RNA in DT40 cells; and accompanied by controls for specificity of this antibody in human cells used for genome wide analysis.

We had performed both DRIP-seq in DT40 and RNA DIP-seq in human samples following *in vitro* RNase H treatment and sequenced them, but had not shown these data in the original figures, as the treatment reduced the amount of recovered material very significantly. We now show the effect of RNase H treatment in the example plots in the new Figures 5 and 6. In addition, all of our R-loop sequencing was performed with RNase A pre-treatment, which will remove chromatin-associated RNAs. In addition, we have added an RNase III control to our *BU-1* DRIP-PCR experiments in Figure 3, discussed above, which confirms that our R-loop signal is RNase III resistant. Together, these controls suggest that the signal we observe in DT40 and human cells is specific for R-loops.

Taking a different tack to identify the source of epigenetic instability, treatment with RNase H1 was shown to abrogate accumulation of BU-1 loss variants (Fig. 3F). The converse experiment, expression of the DNA:RNA hybrid binding domain (HBD) of RNaseH1, caused increased accumulation of loss variants (Fig 4). These results of RNaseH1 and RNaseH1 HBD expression are consistent with the view that RNA/DNA hybrids are in some way responsible for accumulation of loss variants. However, they do not show that the structures form at the BU-1 locus itself, and leave open the possibility RNaseH1 treatment has improved expression or activity of some factor that rescues BU-1 expression.

It is important to note that ectopic RNase H1 expression in *primpol* cells is not rescuing *BU-1* expression but rather the *stability* of *BU-1* expression. The level of *BU-1* expression in the Bu-1^{high} population remains the same. In other words, RNase H1 overexpression does not affect expression of the locus directly but prevents the stochastic instability induced by the presence of the (GAA)₁₀ repeat. We see exactly the same stabilising effect when we delete the (GAA)₁₀ repeat. This is consistent with the observed instability of expression originating from local effects in the gene.

The hybrids formed in the absence of PrimPol are correlated with replication by experiments showing that their abundance increases in S phase, as measured both using the antibody and by metabolic labeling of nascent RNA. As presented, the results of these different analyses (Fig. 5B and Fig. EV6) seem rather different, with the mid-S phase peak in Fig. 5B not evident in Fig. EV6;

and the only clear distinction between wild type and primpol-deficient cells in G2 phase in the latter figure. Is there a real difference? These experiments and results need better explanation.

The original Fig EV6 (now Fig EV4) simply shows cell cycle plots of synchronised and released wild type and *primpol* cultures. However, as discussed in detail above (Reviewer 2, point 2), we have uncovered a significant problem with the slot blot detection of nascent R-loops due to an unanticipated carry-over of RNase A through to the biotinylation step. We have therefore removed the original Figure 5B and replaced it with a completely new experiment (performed in triplicate from the synchronisation to analysis) in which we interrogate the recovered 4-SU-labelled RNA from the *BU-1* locus with qRT-PCR.

The association between R-loops and PrimPol-dependent replication restart genomewide is determined by DRIP-Seq comparing recovery of RNA-DNA hybrids from primpol-deficient and wild type DT40 cells using the S9.6 antibody. The great majority of peaks were common between these two backgrounds, but peak heights were significantly greater in the primpol-deficient cells, suggesting that primpol-deficiency results in a higher steady state level of R-loops, but not induction of new R-loops. This could mean that PrimPol preferentially functions at a subset of genes. This could reflect enrichment of the enzyme's preferred priming site (3'GTCC5': Garcia-Gomez et al MolCell 2013), and even though this site is short it would be useful to score its enrichment/depletion near peaks.

This is an interesting suggestion. However, this is indeed a very short motif and the resolution of peak calling in our DRIP-seq experiments does not allow any firm conclusions to be drawn. Further, we note that while this sequence has been suggested to be preferred, it is certainly seems not to be the only sequence at which PrimPol can initiate repriming and thus the extent to which it dictates repriming *in vivo* remains unclear.

To ask if the peaks identified correlate with possible R-loop formation, enrichment of H-DNA motifs was scored within peaks. H-DNA motifs can correlate with regions with potential to form R-loops, but not enough detail or precedent is presented to rationalize and ground this search. To keep this from seeming like a sleight of hand, more information is required to enable the reader to evaluate the details of the search and especially its stringency. The algorithm needs be stated explicitly and put into context. How long are H-DNA motifs, and how many H-DNA motifs are there in the chicken and human genomes? Has the algorithm used to identify H-DNA motifs been validated in other genomewide analyses? What happens if more (or less) stringent criteria are applied to H-motif identification?

Simple algorithms for detecting H-DNA-forming potential focus only on identifying polypurine or polypyrimidine tracts, but frequently do not allow for interruptions in the tracts causing mismatches in the potential H-DNA structure. They thus only identify a subset of possible motifs. The algorithm we used, 'Triplex' (Hon et al., 2013), takes a dynamic programming-based approach to identify approximate palindromes and as such does allow for mismatches and gaps, using scoring derived from models of the structure of the resulting triplex. The authors of the algorithm validated it using the *E. coli* and human genomes, as described in their paper. Subsequent analyses have used the algorithm in the analysis of prokaryote (Holder et al., 2015), plant (Lexa et al., 2014) and human (Bartholdy et al., 2015) genomes. We have now provided information on the length distribution of H-DNA and G4s identified in the human and chicken genomes (Appendix Fig S10). We have also tried the analysis with simpler algorithms as well as with a number of thresholds for length of H-DNA, including a strict threshold requiring that $\geq 90\%$ of the sequence was comprised of purines/pyrimidines, and found that that the correlation was robust to the choice of any reasonable threshold (i.e. one that does not eliminate almost all of the sequences).

Other information should be provided in order to evaluate the genomewide results: (1) peak width, which determines the potential for overlaps;

We have added graphs of peak widths in the supplementary information. Although there is a small, but significant increase in peak width in the *primpol* samples, this does not account for the increase in peak height. By examining the effect of varying the distance allowed for an overlap to be counted on the number of overlapping peaks called, we shown that this asymptotes at around 1000bp, which is the average size of the input DNA generated by our restriction enzyme cocktail. Importantly, this is also much greater than the observed increase in peak width observed in the *primpol* samples. We have added this data to the supplementary information (new Appendix Figs S8 & 9) and discussed it in the main text.

(2) maps of peak distribution across representative individual genes, identifying sites with potential for formation of R-loops, G4 DNA, etc along the genes - this would substantiate claims of the sort found in the Discussion that PrimPol reprimers at short tandem repeats throughout the genome.

We have now provided example plots from chromosomal regions and specific genes to provide a more direct visual representation of the experimental data. As discussed above, these plots confirm the broad similarity in the pattern of R-loop enrichment in wild type and *primpol* mutants and, because the data are normalised to a spike-in control, allow the increase in R-loop signal at peaks in *primpol* to be clearly seen. Importantly, it also shows that not every site capable of G4 and H-DNA formation, as detected by our algorithms, is associated with R-loops but that, in general, R-loop peaks that are near identified HDNA or G4-forming sequences are higher in *primpol* cells, as confirmed by the genome-wide analysis.

Very high statistical significance is reported in both DT40 chicken B cells and the human BOBSC stem cell line in Figure 6, but the labels in the figure and the details of the analysis are somewhat unclear. Is the presence of a single H-DNA motif in a long gene sufficient for it to score in the statistical analysis?

Yes, it is. We assume the reviewer has concerns that we should have normalised for gene length. We are uncertain exactly which part of this figure the reviewer might have concerns with so we address this issue in all parts that mention secondary structures. In the original Figs 6D,F,K and M (new Figs 5F & H and 6G & H) we use the same subset of genes in both wild type and *primpol* so length cancels out. In original Fig 6C,E,I and L (new Figs 5E & G and 6E & F), the same subset of genes is not used across the comparisons (though the sets are ~80% identical). However, we tested whether genes with peaks were significantly different in length between wild type and *primpol* and they were not, as such normalisation would not affect the results shown here, especially given their degree of significance.

We also noticed a figure preparation error that left the p-values in the original panel C duplicated in panels E, J and L. This has been corrected in the revised figures but alters none of the conclusions.

Major comments:

1. Include controls for S9.6 antibody specificity in DT40 and human BOBSC stem cells.

See discussion above. We had performed the key controls now show them explicitly both for DRIP-PCR and genome wide analyses.

2. DRIP-Seq peak widths must be presented to understand the significance of overlaps. It would be useful to see how results change as peak widths are narrowed; and with changes in stringency of the H-motif algorithm.

See discussion above. We have added peak width data to the revised manuscript (Appendix Figs S8 & 9) and discussed the significance in the main text.

The Introduction and Discussion over-emphasize short repetitive tracts, considering that only one specific example is shown, the GAA repeat cloned into the BU-1 locus.

We have modified the introduction and discussion to clarify that the studies in *BU-1* employ (GAA)₁₀ as a prototypical short repetitive tract with structure-forming potential. At the same time, we wish to make it clear that such low complexity sequences with structure forming potential are common. We believe that our findings reflect a principle that is generalisable to other short, structure-forming sequences.

Minor comments:

Discussion: "We now show that the repriming function of PrimPol is frequently deployed at short tandem repeats throughout the genome...." This has not been shown and must be rephrased. Demonstrating this could be done with more detailed presentation of the genomewide analysis, which would be appropriate and interesting.

We agree the phrasing of this sentence is an overstatement and have altered it. While our genome wide analysis supports our hypothesis that loss of PrimPol increases S phase R-loop formation in the vicinity of secondary structure-forming sequences, the resolution of the techniques employed does not currently permit a more detailed analysis.

Fig. 3E: Overlay cell cycle profile on data

The cell cycle data is already provided though the DNA content stain (X-axis, Hoechst 33342), and we have made this clearer by showing the DNA content histogram aligned to the X-axis.

Figures 2 and 4 might be presented as supplementary data rather than in the main text. This would give space for deeper analysis of the genomewide results.

We have reorganised the figures so that the genome-wide analysis now occupies Figures 5 and 6.

Fig. EV6: Numbers in flow cytometry output quadrants should be large enough to read.

We have modified the figure.

References

Bartholdy B, Mukhopadhyay R, Lajugie J, Aladjem MI, Bouhassira EE (2015) Allele-specific analysis of DNA replication origins in mammalian cells. *Nat Commun* 6: 7051

- Dona F, Houseley J (2014) Unexpected DNA loss mediated by the DNA binding activity of ribonuclease A. *PLoS One* 9: e115008
- Felsenfeld G, Sandeen G, Vonhippel PH (1963) The Destabilizing Effect of Ribonuclease on the Helical DNA Structure. *Proc Natl Acad Sci U S A* 50: 644-51
- Ginno PA, Lott PL, Christensen HC, Korf I, Chedin F (2012) R-loop formation is a distinctive characteristic of unmethylated human CpG island promoters. *Mol Cell* 45: 814-25
- Grabczyk E, Fishman MC (1995) A long purine-pyrimidine homopolymer acts as a transcriptional diode. *J Biol Chem* 270: 1791-7
- Grabczyk E, Mancuso M, Sammarco MC (2007) A persistent RNA.DNA hybrid formed by transcription of the Friedreich ataxia triplet repeat in live bacteria, and by T7 RNAP in vitro. *Nucleic Acids Res* 35: 5351-9
- Grabczyk E, Usdin K (2000) The GAA*TTC triplet repeat expanded in Friedreich's ataxia impedes transcription elongation by T7 RNA polymerase in a length and supercoil dependent manner. *Nucleic Acids Res* 28: 2815-22
- Groh M, Gromak N (2014) Out of balance: R-loops in human disease. *PLoS Genet* 10: e1004630
- Guilbaud G, Murat P, Recolin B, Campbell BC, Maiter A, Sale JE, Balasubramanian S (2017) Local epigenetic reprogramming induced by G-quadruplex ligands. *Nat Chem* 9: 1110-1117
- Holder IT, Wagner S, Xiong P, Sinn M, Frickey T, Meyer A, Hartig JS (2015) Intrastrand triplex DNA repeats in bacteria: a source of genomic instability. *Nucleic Acids Res* 43: 10126-42
- Hon J, Martinek T, Rajdl K, Lexa M (2013) Triplex: an R/Bioconductor package for identification and visualization of potential intramolecular triplex patterns in DNA sequences. *Bioinformatics* 29: 1900-1
- Kobayashi K, Guillian TA, Tsuda M, Yamamoto J, Bailey LJ, Iwai S, Takeda S, Doherty AJ, Hirota K (2016) Repriming by PrimPol is critical for DNA replication restart downstream of lesions and chain-terminating nucleosides. *Cell Cycle* 15: 1997-2008
- Lexa M, Kejnovsky E, Steflava P, Konvalinova H, Vorlickova M, Vyskot B (2014) Quadruplex-forming sequences occupy discrete regions inside plant LTR retrotransposons. *Nucleic Acids Res* 42: 968-78
- Mouron S, Rodriguez-Acebes S, Martinez-Jimenez MI, Garcia-Gomez S, Chocron S, Blanco L, Mendez J (2013) Repriming of DNA synthesis at stalled replication forks by human PrimPol. *Nat Struct Mol Biol* 20: 1383-9
- Papadopoulou C, Guilbaud G, Schiavone D, Sale JE (2015) Nucleotide Pool Depletion Induces G-Quadruplex-Dependent Perturbation of Gene Expression. *Cell Rep* 13: 2491-2503
- Punga T, Buhler M (2010) Long intronic GAA repeats causing Friedreich ataxia impede transcription elongation. *EMBO Mol Med* 2: 120-9
- Roy D, Yu K, Lieber MR (2008) Mechanism of R-loop formation at immunoglobulin class switch sequences. *Mol Cell Biol* 28: 50-60
- Sanz LA, Hartono SR, Lim YW, Steyaert S, Rajpurkar A, Ginno PA, Xu X, Chedin F (2016) Prevalent, Dynamic, and Conserved R-Loop Structures Associate with Specific Epigenomic Signatures in Mammals. *Mol Cell* 63: 167-78
- Sarkies P, Murat P, Phillips LG, Patel KJ, Balasubramanian S, Sale JE (2012) FANCDJ coordinates two pathways that maintain epigenetic stability at G-quadruplex DNA. *Nucleic Acids Res* 40: 1485-98

Schiavone D, Guilbaud G, Murat P, Papadopoulou C, Sarkies P, Prioleau MN, Balasubramanian S, Sale JE (2014) Determinants of G quadruplex-induced epigenetic instability in REV1-deficient cells. *EMBO J* 33: 2507-20

Schiavone D, Jozwiakowski SK, Romanello M, Guilbaud G, Guillian TA, Bailey LJ, Sale JE, Doherty AJ (2016) PrimPol Is Required for Replicative Tolerance of G Quadruplexes in Vertebrate Cells. *Mol Cell* 61: 161-9

Taylor MRG, Yeeles JTP (2018) The Initial Response of a Eukaryotic Replisome to DNA Damage. *Mol Cell* 70: 1067-1080 e12

2nd Editorial Decision

25th October 2018

Thank you for submitting your revised manuscript for our consideration. It has now been seen once more by two of the original reviewers, whose comments are copied below. I am pleased to inform you that with both of them largely satisfied with the revisions and improvements to the paper, we shall be happy to publish this work in The EMBO Journal, following some remaining minor modifications.

As you will see, referee 3 still retains some concerns regarding the use of the S9.6 antibody and the associated conclusions. Please consider them carefully and provide a detailed response to them; in the manuscript itself, I feel that at least some clear further explanation/clarification would be warranted, and possibly moderating of some claims/discussion of potential caveats.

Please return the modified text file (and, if applicable, modified figure files) simply via email to me. Once we will have received these final files and synopsis items, I hope we should be able to swiftly proceed with formal acceptance and publication of the study!

REFEREE REPORTS.

Referee #1 (Report for Author)

The authors addressed all relevant concerns and I recommend the manuscript for publication.

Referee #3 (Report for Author)

The authors have revised the manuscript in response to previous comments. The revisions have improved the manuscript, particularly the recognition that RNase A binding to DNA was creating an artefact in the previous cell cycle analyses. However, some questions about signal detection and S9.6 antibody specificity are only partially resolved.

Appendix Figure S2 is presented to address some of these concerns. This figure presents an analysis of RNaseH and RNaseIII sensitivity of the BU-1 locus. This experimental design is a little circular, since this is the locus being investigated, and use of a different locus at which R-loop formation has been independently verified would be a better positive control. Without that, the terminus can be seen as something of an internal control for what constitutes a target for RNaseH digestion. The experiments shown analyze the response of different regions of BU-1 locus to digestion with RNaseH and RNaseIII. RNaseH treatment reduces signal efficiently at the repeat, but less well at the terminus, where reduction is in the range of 2- to 2.5-fold; p0.1. What accounts for the remaining signal, which by itself appears to be significant relative to background from the gene body? The experiment does control for the obvious possibility that dsRNA is contributing to the signal, by examining sensitivity to RNaseIII, but the terminus signal was not reduced by RNaseIII digestion. The signal could be caused by something other than an RNA-DNA hybrid; or it can be seen as a

reminder that RNaseH-resistance depends on the parameters of the assay, especially RNaseH concentration and incubation time and temperature, in which case it raises a caveat about use of RNaseH digestion as the control for S9.6 antibody specificity, here and elsewhere in the manuscript.

Uncertainty regarding S9.6 specificity also raises questions about the signal from the gene body. Sanz et al. used the S9.6 antibody in the report that classified about 25% of genes as "sticky", which the authors cite as a potential explanation for the puzzling signal they observe. If the S9.6 antibody is less specific than claimed, then there may be a different explanation for signal from gene body.

Final Revision - authors' response

29th October 2018

EMBOJ: Manuscript EMBOJ-2018-99793R2

Response to comments (in blue):

Referee #1 (Report for Author)

The authors addressed all relevant concerns and I recommend the manuscript for publication.

Referee #3 (Report for Author)

The authors have revised the manuscript in response to previous comments. The revisions have improved the manuscript, particularly the recognition that RNase A binding to DNA was creating an artefact in the previous cell cycle analyses. However, some questions about signal detection and S9.6 antibody specificity are only partially resolved.

Appendix Figure S2 is presented to address some of these concerns. This figure presents an analysis of RNaseH and RNaseIII sensitivity of the BU-1 locus. This experimental design is a little circular, since this is the locus being investigated, and use of a different locus at which R-loop formation has been independently verified would be a better positive control.

S9.6 remains the most widely used single tool for the analysis of R-loops. However, as for all antibodies, its use comes with caveats, which have been discussed extensively (Phillips *et al.*, 2013; Hartono *et al.*, 2018; Vanoosthuyse, 2018). The sensitivity of an S9.6 signal to RNaseH and its resistance to RNaseIII in samples pre-treated also with RNaseA does provide significant assurance that R-loops are being detected. We believe that we have performed all the currently accepted controls needed to support our conclusion that the signals we detect with S9.6 are R-loops. Since this work presents the first comprehensive analysis of R-loop formation in DT40 cells, we are not clear what constitutes an independently verified positive control. We do show that RNaseH treatment massively reduces the amount of immunoprecipitated material in the DRIP-seq experiments, as shown in Figures 5A and 6A. Importantly, our study also develops and deploys an S9.6-independent method to detect R-loop synthesis (Figure 4D-G), which supports our model. Further, we show clear phenotypic consequences in terms of *BU-1* expression stability when we overexpress RNaseH1 in DT40 cells.

Without that, the terminus can be seen as something of an internal control for what constitutes a target for RNaseH digestion. The experiments shown analyze the response of different regions of BU-1 locus to digestion with RNaseH and RNaseIII. RNaseH treatment reduces signal efficiently at the repeat, but less well at the terminus, where reduction is in the range of 2- to 2.5-fold; $p < 0.1$. What

accounts for the remaining signal, which by itself appears to be significant relative to background from the gene body?

We believe that the remaining signal at the 3' end of the locus following *in vitro* RNaseH digestion likely reflects incomplete digestion of the extensive RNA:DNA hybrids at the 3' end of the locus, under the conditions we employed, combined with the high sensitivity of the subsequent qPCR detection. [The referee's comments about the p value being 0.1 prompted us to check the legend of Figure S2 in the appendix as a single asterisk should indicated $p < 0.05$. There was indeed a typo here, which is now corrected. The reduction is significant at $p < 0.05$]. We note that the 2 – 2.5-fold reduction of the 3' S9.6 signal we seen in *BU-1* following RNaseH-treatment is similar to that reported for the equivalent signal detected in the human beta-actin locus (Skourti-Stathaki *et al.*, 2014).

The experiment does control for the obvious possibility that dsRNA is contributing to the signal, by examining sensitivity to RNaseIII, but the terminus signal was not reduced by RNaseIII digestion. The signal could be caused by something other than an RNA-DNA hybrid; or it can be seen as a reminder that RNaseH-resistance depends on the parameters of the assay, especially RNaseH concentration and incubation time and temperature, in which case it raises a caveat about use of RNaseH digestion as the control for S9.6 antibody specificity, here and elsewhere in the manuscript.

Recently, a concern has been raised (Vanoosthuysse, 2018) non-B DNA structures associated with R-loops can be recognised by the S9.6 antibody and that this could account for RNaseH-resistant signals. As the reviewer points out we control for the possibility that dsRNAs are contributing to the signal and, in addition, all our samples are pretreated with RNaseA, which will remove free RNA that could form dsRNA species during sample preparation. While we significantly reduce the 3' S9.6 signal with RNaseH, the same treatment abrogates the signal in the gene body, which is the focus of this study.

It was because of some concerns over the specificity of S9.6 that we developed an S9.6-independent method to monitor nascent R-loops formation. We use this approach to show R-loop formation in *BU-1* at the time the locus is replication and, importantly for this argument, show that this signal is completely RNaseH sensitive. Additionally, all the experiments in *BU-1* are genetically controlled. We therefore do not believe that this concern impacts the conclusion of the study. We have added reference to the concerns and caveats around S9.6 use in the main text.

Uncertainty regarding S9.6 specificity also raises questions about the signal from the gene body. Sanz *et al.* used the S9.6 antibody in the report that classified about 25% of genes as "sticky", which the authors cite as a potential explanation for the puzzling signal they observe. If the S9.6 antibody is less specific than claimed, then there may be a different explanation for signal from gene body.

The observation that the R-loop signal extends both sides of the structure-forming sequence is clear. We offer the above hypothesis as one potential explanation. However, while we have clearly demonstrated that repriming failure leads to R-loop synthesis during replication, we are certain that we do not yet have a full picture of what happens when RNAPII does indeed load onto the ssDNA created by helicase-polymerase uncoupling events.

References

- Hartono SR, Malapert A, Legros P, Bernard P, Chédin F, Vanoosthuyse V (2018) The Affinity of the S9.6 Antibody for Double-Stranded RNAs Impacts the Accurate Mapping of R-Loops in Fission Yeast. *J Mol Biol*, **430**: 272–284
- Phillips DD, Garboczi DN, Singh K, Hu Z, Leppla SH, Leysath CE (2013) The sub-nanomolar binding of DNA-RNA hybrids by the single-chain Fv fragment of antibody S9.6. *J Mol Recognit*, **26**: 376–381
- Skourti-Stathaki K, Kamieniarz-Gdula K, Proudfoot NJ (2014) R-loops induce repressive chromatin marks over mammalian gene terminators. *Nature*, **516**: 436–439
- Vanoosthuyse V (2018) Strengths and Weaknesses of the Current Strategies to Map and Characterize R-Loops. *Noncoding RNA*, **4**:

Accepted

6th November 2018

Thank you for submitting your final revised manuscript files for our consideration. Having now carefully gone through them, I am pleased to inform you that we have now accepted the study for publication in The EMBO Journal!

Corresponding Author Name: Julian Sale

Journal Submitted to: EMBO JOURNAL

Manuscript Number: EMBOJ-2018-99793R